# A retinoraphe projection regulates serotonergic activity and looming-evoked defensive behaviour

Lu Huang[1,2], Tifei Yuan[3], Minjie Tan[1,2], Yue Xi[1,2], Yu Hu[1,2], Qian Tao[4], Zhikai Zhao[1,2], Jiajun Zheng[1,2], Yushui Han[1,2], Fuqiang Xu[5], Minmin Luo[6], Patricia J. Sollars[7], Mingliang Pu[8], Gary E. Pickard[7,9], Kwok-Fai So[1,2,10,11] & Chaoran Ren[1,2,11]

Animals promote their survival by avoiding rapidly approaching objects that indicate threats. In mice, looming-evoked defensive responses are triggered by the superior colliculus (SC) which receives direct retinal inputs. However, the specific neural circuits that begin in the retina and mediate this important behaviour remain unclear. Here we identify a subset of retinal ganglion cells (RGCs) that controls mouse looming-evoked defensive responses through axonal collaterals to the dorsal raphe nucleus (DRN) and SC. Looming signals transmitted by DRN-projecting RGCs activate DRN GABAergic neurons that in turn inhibit serotoninergic neurons. Moreover, activation of DRN serotoninergic neurons reduces looming-evoked defensive behaviours. Thus, a dedicated population of RGCs signals rapidly approaching visual threats and their input to the DRN controls a serotonergic self-gating mechanism that regulates innate defensive responses. Our study provides new insights into how the DRN and SC work in concert to extract and translate visual threats into defensive behavioural responses.

[1] Guangdong-Hongkong-Macau Institute of CNS Regeneration, Ministry of Education CNS Regeneration Collaborative Joint Laboratory, Jinan University, Guangzhou 510632, China. [2] Guangdong key Laboratory of Brain Function and Diseases, Jinan University, Guangzhou 510632, China. [3] School of Psychology, Nanjing Normal University, Nanjing 210097, China. [4] Psychology Department, School of Medicine, Jinan University, Guangzhou 510632, China. [5] State Key Laboratory of Magnetic Resonance and Atomic and Molecular Physics, Wuhan Institute of Physics and Mathematics, Chinese Academy of Sciences, Wuhan 430071, China. [6] National Institute of Biological Sciences, Zhongguancun Life Science, Park 7 Science Park Road, Beijing 102206, China. [7] School of Veterinary Medicine and Biomedical Sciences, University of Nebraska, Lincoln, Nebraska 68583, USA. [8] Department of Anatomy, School of Basic Medical Sciences, Peking University, Beijing 100191, China. [9] Department of Ophthalmology and Visual Sciences, University of Nebraska Medical Center, Omaha, Nebraska 68198, USA. [10] Department of Ophthalmology and State Key Laboratory of Brain and Cognitive Sciences, The University of Hong Kong, Hong Kong, China. [11] Co-innovation Center of Neuroregeneration, Nantong University, Nantong 226001, China. Correspondence and requests for materials should be addressed to K.-F.S. (email: hrmaskf@hku.hk) or to C.R. (email: tchaoran@jnu.edu.cn).

Defensive behaviour in response to rapidly approaching objects is an innate self-defence response remarkably conserved across species, including humans[1–8]. Simulations of a looming object that mimic an approaching aerial predator can initiate a rapid escape response and retinal projections to the midbrain optic tectum or the mammalian superior colliculus (SC) provide the sensory input that initiates rapid defence responses[9–12]. The SC in turn informs the amygdala where fear responses are assigned to this specific sensory experience[13,14]. The neural circuits beginning in the retina that regulate looming-evoked defensive behaviour are not well understood.

The ascending serotonergic system derived from the midbrain dorsal raphe nucleus (DRN) may influence the expression of defensive responses through activation of a diverse set of serotonin (5-HT) receptor subtypes distributed in fear/threat-related regions, including the amygdala and prefrontal cortex[15–20]. Revealing the structure and function of the precise circuit related to the DRN is important for the understanding of how the serotonergic system exerts its influence upon defensive responses. The DRN in several species including primates receives retinal afferents and DRN-projecting retinal ganglion cells (RGCs) regulate serotonergic signalling in the brain[21,22]. It is worth noting that in the Mongolian gerbil DRN-projecting RGCs also send collateral branches to the SC[23]. Therefore, if the DRN received looming-related signals transmitted from RGCs with branching axons innervating the SC, it might be in a position to modulate looming-evoked behaviour via its extensive 5-HT projections.

In this study, by combining conventional neurotracer and transneuronal rabies virus tracing techniques, we identified a retinoraphe projection in the mouse with DRN-projecting RGCs also sending branching axons to the SC. In the DRN these RGCs selectively innervate GABA neurons, which in turn modulate DRN 5-HT neuron activity. The role of DRN/SC-projecting RGCs in the regulation of looming-evoked defensive responses was examined using an array of brain circuit interrogation tools, including immunotoxin-based RGC ablation, c-Fos activity mapping, fibre photometry, chemogenetics and optogenetics. We report that DRN/SC-projecting RGCs are necessary for looming-induced defensive responses and that DRN 5-HT neuron activity regulates the circuits that control looming-evoked behaviour.

## Results

**RGCs innervate the mouse DRN and SC.** Retrograde tracing was conducted to determine if a retinoraphe projection exists in the mouse. To deliver retrograde tracer into the DRN while avoiding the retinorecipient SC, an angled approach was used and the DRN was reached by passing the electrode through the inferior colliculus (Fig. 1a); each animal ($n = 8$) received a single injection of 200 nl of cholera toxin B subunit (CTB) conjugated to Alexa Fluor 488 (CTB-488). Injections centred in the DRN with no contamination of the SC (Fig. 1b and Supplementary Fig. 1a–e) labelled approximately 600 RGCs uniformly distributed across the retina ($573 \pm 21$ RGCs per retina, $n = 16$; Fig. 1c). The results indicate that retrograde transport of CTB-488 is an effective technique for labelling DRN-projecting RGCs in the mouse in contrast to intraocular CTB-488 injections which do not adequately label the axons of these RGCs in the DRN[24].

We next asked whether RGCs in the mouse send axon collaterals to the DRN and SC, similar to those in the gerbil[23]. Using CTB tracer injections into the DRN (CTB-488) and SC (CTB-594) to retrogradely label RGCs ($n = 4$ animals) (Supplementary Fig. 2a), we observed that approximately 90%

of CTB-488-labelled DRN-projecting RGCs were double-labelled (581 of 652 cells analysed), indicating that individual RGCs in the mouse send branching axons to the DRN and SC (Supplementary Fig. 2b–e).

In an attempt to classify DRN-projecting RGCs we used three well-known markers of RGCs: melanopsin, cocaine and amphetamine-regulated transcript (CART), and SMI-32. No CTB-labelled DRN-projecting RGCs were immunopositive for melanopsin, CART or SMI-32 (Supplementary Fig. 2f–n). Next we sought to classify DRN-projecting RGCs based on their pattern of dendritic arborization. CTB-labelled DRN-projecting RGCs in retinal whole mounts maintained *in vitro* were randomly selected and targeted for neurobiotin intracellular filling (Fig. 1d). DRN-projecting RGCs have a medium-sized round soma (diameter $= 17.3 \pm 2.6 \mu m$; $n = 472$ from 6 retinas) with 3–5 primary dendrites that were generally thick and smooth (Fig. 1d–g and Supplementary Fig. 3a–f). The dendritic field of DRN-projecting RGCs covers approximately $0.043 \pm 0.004 \text{ mm}^2$ and is asymmetrical; the average dendritic field size and soma diameter did not vary significantly with retinal eccentricity (Supplementary Fig. 3g,h). In z-stack reconstructions of filled DRN-projecting RGCs, dendritic processes ramified in both sublamina *a* and *b* of the inner plexiform layer (Supplementary Fig. 3a–f).

**DRN-projecting RGCs selectively innervate DRN GABA neurons.** To exclude the possibility that DRN CTB injections inadvertently labelled RGC fibres passing through rather than synapsing in the DRN[25], and further to determine if DRN-projecting RGCs specifically target one or both of the two major DRN cell types (5-HT and GABA), a three-virus tracing technique was used in combination with Cre-loxP gene-expression[26]. The modified rabies virus SAD-ΔG-DsRed (EnvA) was pseudotyped with the avian sarcoma leucosis virus envelope protein (EnvA), limiting the rabies virus to infect only those neurons that express TVA, a cognate receptor of EnvA. Additionally, the rabies glycoprotein (RG) gene required for transneuronal spread beyond the initially infected neurons (starter cells) was replaced with the coding sequence of a red fluorescent protein, DsRed (Fig. 2a). Two Cre-dependent AAV helper virus recombinants, AAV-DIO-EGFP-TVA and AAV-DIO-RVG, were infused into the DRN of two strains of mice expressing Cre-recombinase under the transcriptional control of: (1) the serotonin transporter promotor (Sert-Cre mice); or (2) the vesicular GABA transporter promotor (vGAT-Cre mice) to selectively infect DRN 5-HT or GABA neurons. After allowing 3 weeks for the expression of EGFP-TVA and RG in the DRN, the rabies virus ΔRG-DsRed (EnvA) was injected into the same area (Fig. 2b). One week later retinas and brains were examined for neurons labelled by the rabies virus.

The expression of Cre-dependent TVA-EGFP in the DRN was confirmed to be highly specific: 96.4% of TVA-EGFP labelled cells were tryptophan hydroxylase (TPH)-immunopositive in Sert-Cre animals (648 of 672 cells analysed) and 97.7% of TVA-EGFP labelled cells in the DRN were non-TPH-immunopositive cells in vGAT-Cre animals (545 of 558 cells analysed) (Fig. 2c,d). Starter cells were co-labelled with TVA-EGFP and SAD-ΔG-DsRed (EnvA) and were restricted to the DRN (Fig. 2c,d and Supplementary Fig. 4a). In Sert-Cre animals there were numerous rabies virus labelled (DsRed-positive) neurons in the DRN which did not express TVA-EGFP, indicating that these cells were retrogradely labelled from DRN 5-HT starter cells and are therefore local inputs; the fact that DRN GABA interneurons innervate 5-HT neurons is well-documented[27] (Fig. 2c,d). As described previously, we confirmed that DRN 5-HT and GABA

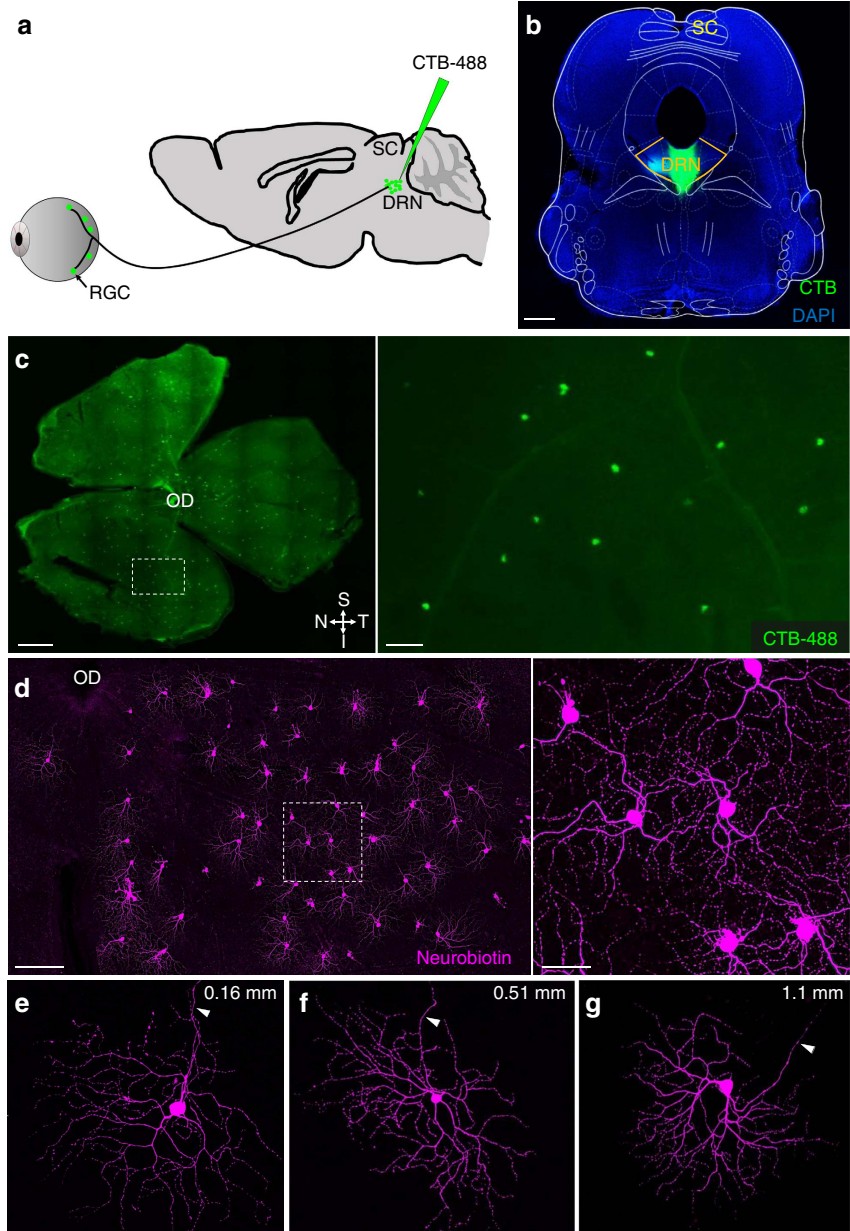

**Figure 1 | Retinal ganglion cells (RGCs) innervate the mouse DRN.** (**a**) Angled trajectory used to eject cholera toxin B subunit Alexa Fluor 488 (CTB-488) into the dorsal raphe nucleus (DRN) while avoiding the superior colliculus (SC). (**b**) Image of a CTB-488 injection site in the DRN. Note absence of CTB-488 in the SC; yellow lines demarcate borders of the DRN. (**c**) Left: retinal whole mount illustrating retrogradely labelled DRN-projecting RGCs. Retina orientation: S = superior, I = inferior, N = nasal and T = temporal. Right: region defined in white box viewed under higher magnification. (**d**) Left: intracellularly filled DRN-projecting RGCs in a retinal whole mount. Right: region defined in white box viewed under higher magnification. (**e–g**) Intracellularly filled DRN-projecting RGCs located at different retinal eccentricities relative to optic disc (OD). Arrowheads point to axons. Numbers in the upper right corner indicate distance between soma and optic disc. Scale bars: (**b**,**c**-left) 500 μm; (**c**-right, **d**-right and **e**) 50 μm; (**d**-left) 100 μm.

neurons receive direct innervation from dozens of brain areas, including the central amygdala and habenula (Supplementary Fig. 4b)[27–29]. Labelled RGCs were restricted almost without exception to the retinas of vGAT-Cre mice: 118 labelled RGCs in eight vGAT-Cre mouse retinas compared to only two labelled RGCs in 16 Sert-Cre mouse retinas (Fig. 2e). The dendritic morphology and distribution pattern of labelled RGCs in vGAT-Cre mice matched that of CTB-labelled DRN-projecting RGCs filled with neurobiotin (Fig. 2f,g and Supplementary Fig. 4c). The results from these selective rabies virus transneuronal tracing experiments provide direct evidence that RGCs synapse on DRN GABA neurons.

**Looming-evoked response requires DRN-projecting RGCs.** RGCs that respond to looming visual stimuli send signals to the SC to initiate defensive responses. Since DRN-projecting RGCs also innervate the SC, we asked whether these RGCs play a role in mediating responses to looming stimuli. Using extracellular recording in retinal whole mounts maintained *in vitro*, we examined responses of DRN-projecting RGCs to a looming stimulus (a dark disk on a grey background that rapidly expanded from 2° visual angle to 20° visual angle in 250 ms (ref. 7)). All of the recorded CTB-labelled DRN-projecting RGCs ($n = 11$) showed a transient increase in their firing rate at the onset of the looming stimulus (Fig. 3a). Some RGCs ($n = 4$) were filled with

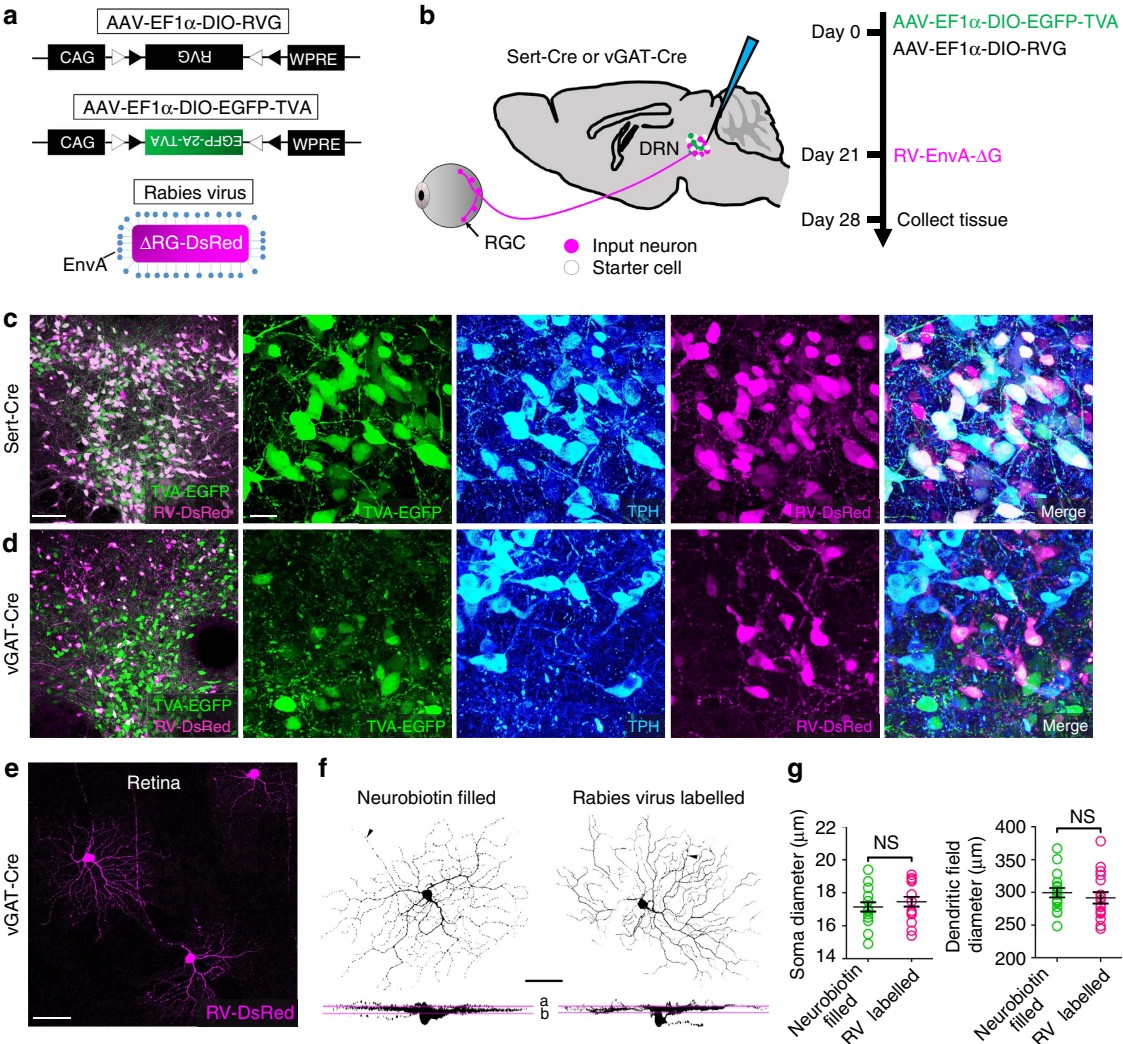

**Figure 2 | DRN-projecting RGCs selectively innervate DRN GABA neurons.** (**a**) Design of AAV-DIO-RVG, AAV-DIO-EGFP-TVA and SAD-ΔG-DsRed (EnvA). (**b**) Experimental design of virus tracing in Sert-Cre and vGAT-Cre mice. (**c**) Left, Sert-Cre mouse DRN showing the location of starter cells (white). Right, a DRN section viewed under higher magnification showing from left to right: green for TVA-EGFP; blue for anti-TPH immunostaining; magenta for rabies-DsRed; and merged image showing triple-labelled cells. (**d**) Left, vGAT-Cre mouse DRN showing the location of starter cells (white). Right, a DRN section viewed under higher magnification showing from left to right: green for TVA-EGFP; blue for anti-TPH immunostaining; magenta for rabies-DsRed and merged image. (**e**) DRN-projecting RGCs in a retinal whole mount labelled with rabies virus in a vGAT-Cre mouse. (**f**) Dendritic morphology of a DRN-projecting RGC filled with neurobiotin in a wild-type mouse (left) and a DRN-projecting RGC retrogradely labelled by rabies virus in a vGAT-Cre mouse (right). Note that both cells have asymmetric dendritic fields and bistratified dendritic arbors. **a,b**: sublaminas a and b of the inner plexiform layer (IPL). (**g**) Comparison of soma and dendritic field diameters between DRN-projecting RGCs filled with neurobiotin or labelled by rabies virus. ns = no significant difference; Scale bars: (**c**)-left 100 μm; (**c**)-right 20 μm; (**e,f**) 50 μm. Data presented as mean ± s.e.m.

neurobiotin confirming that their dendritic morphology was consistent with previously filled DRN-projecting RGCs (Fig. 3a). We next wanted to test whether these RGCs are required for looming-evoked behavioural responses. First we confirmed that mouse DRN-projecting RGCs labelled with CTB could be selectively ablated by intraocular injection of the ribosomal toxin saporin (SAP) conjugated to an antibody generated against CTB (anti-CTB-SAP) as we reported previously in the gerbil[21]. This tool is effective because CTB retrogradely transported to the RGC soma is delivered to the cell surface via transcytosis[30], providing anti-CTB-SAP access to CTB[31]. Once bound, the immunotoxin is internalized to exert its effect as depicted in Fig. 3b. In retinas treated with anti-CTB-SAP the number of CTB-labelled DRN-projecting RGCs was reduced approximately 90% compared to vehicle-injected animals (VEH 562.74 ± 20.0, n = 9 versus anti-CTB-SAP 68.2 ± 16.6, n = 8; P < 0.0001) (Fig. 3c–e). These results provide the basis for functional studies examining the role of DRN-projecting RGCs in looming-evoked behaviour.

To determine if DRN-projecting RGCs are required for looming-evoked escape responses, we used a behavioural assay of a rapidly expanding dark disk stimulus similar to that used for the *in vitro* experiment (looming stimulus −2° of visual angle expanded to 20° in 250 ms) presented overhead 15 times over 10.75 s to evoke an escape response in which mice flee-to-shelter[7] (Fig. 3f). The DRN of all animals in these experiments was injected with CTB and all animals also received intraocular injections of: (1) phosphate-buffered saline (PBS) (VEH, n = 27); (2) the immunotoxin, anti-CTB-SAP (CTB-SAP, n = 10); or (3) the immunotoxin, anti-melanopsin conjugated to saporin[21] (anti-Mel-SAP, n = 8).

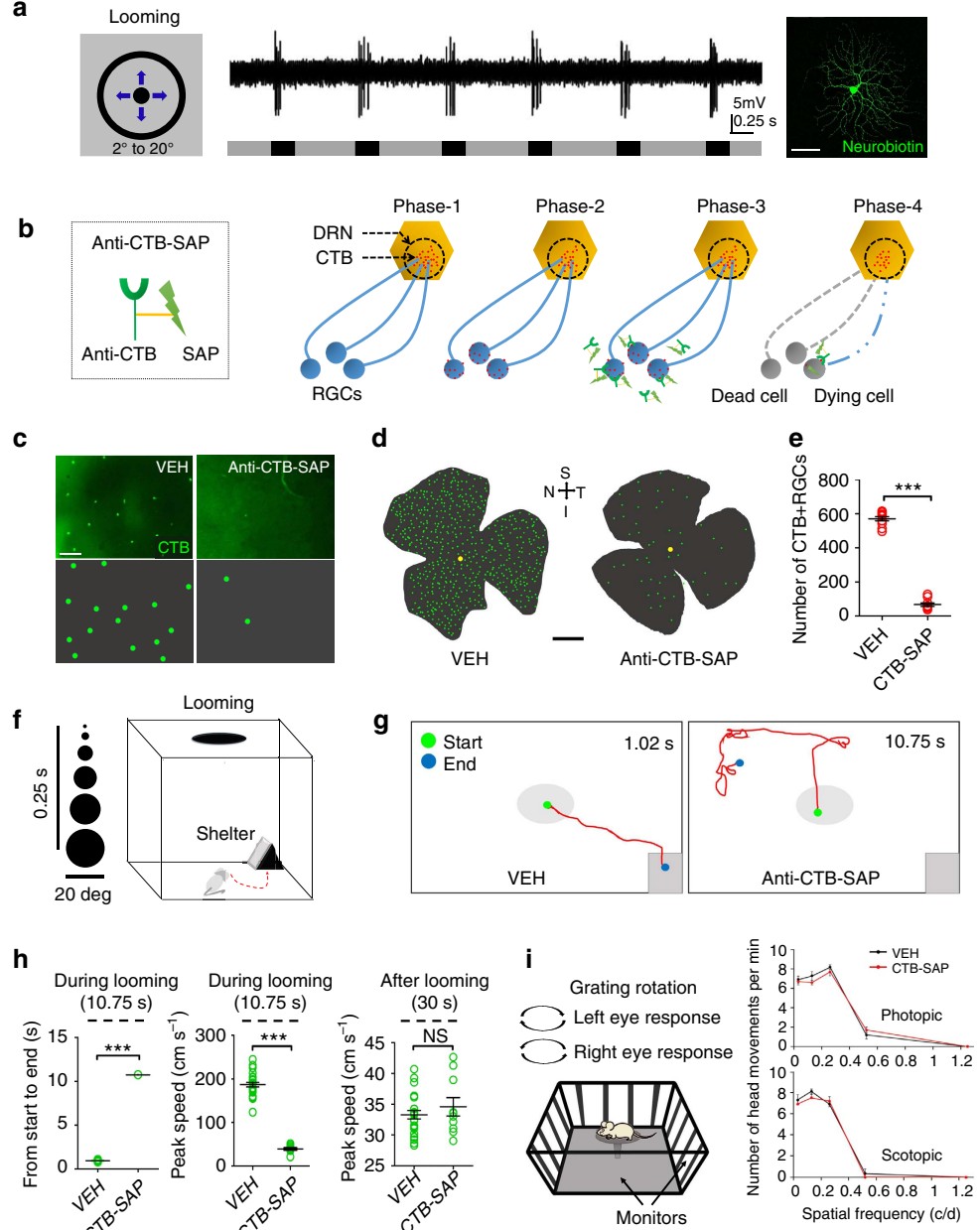

**Figure 3 | DRN-projecting RGCs are necessary for looming-evoked defensive responses.** (**a**) Responses of a DRN-projecting RGC to looming stimulation. Left, an expanding dark disk on a grey background used as a looming stimulus; Centre, example of physiological responses of a DRN-projecting RGC to looming stimulus; Right, DRN-projecting RGC filled with neurobiotin after recording physiological responses to looming stimulus. (**b**) Design of immunotoxin anti-CTB-SAP: IgG antibody generated against CTB conjugated to saporin (SAP). Immunotoxin binds to CTB (retrogradely transported to the soma and delivered to the cell surface via transcytosis) and enters the cell through endocytosis. SAP kills the cell by inactivating ribosomes. (**c**) Examples of CTB-labelled DRN-projecting RGCs in retinal whole mounts after intraocular PBS (VEH) or anti-CTB-SAP (2 µg per eye) injection. CTB-labelled DRN-projecting RGCs are enhanced in the lower panels. (**d**) Distribution of CTB-labelled DRN-projecting RGCs (enhanced) in retinal whole mounts 14 days after intraocular vehicle (left) or anti-CTB-SAP (right) injections. S: Superior, I: Inferior, N: Nasal and T: Temporal. (**e**) Quantification of DRN-projecting RGCs in vehicle (VEH) and anti-CTB-SAP (CTB-SAP)-treated animals. ($n = 8$ retinas per group), one-way ANOVA; ***$P < 0.0001$. (**f**) Schematic of looming animation and testing arena. (**g**) Representative traces of animal movement during looming stimulation (10.75 s) in VEH and anti-CTB-SAP groups; time in corner is duration of trace from start to end. (**h**) Duration from start to end points, peak speed during and after looming stimulation in VEH and anti-CTB-SAP (CTB-SAP)-treated animals. One-way ANOVA; ***$P < 0.0001$; ns = no significant difference. (**i**) The optomotor response of animals treated with VEH or anti-CTB-SAP (CTB-SAP) tested under photopic and scotopic conditions; the responses in both animal groups under both conditions were similar. Scale bars: (**a,c**) 50 µm; (**d**) 1 mm. Data presented as mean ± s.e.m.

In response to the looming stimulus, a rapid escape to the shelter was observed in >90% of the VEH animals (25 of 27) (Fig. 3g and Supplementary Movie 1). Two animals froze for a prolonged period in response to the looming stimulus and these animals were not included in the kinetic analysis (Fig. 3h). The looming stimulus did not evoke escape responses in the anti-CTB-SAP-treated animals; these mice continued to explore the arena during and after the looming stimulus (Fig. 3g,h and

Supplementary Movie 1). Based on the anatomical analyses described above, intraocular injection of anti-CTB-SAP would have killed approximately 90% of the CTB-labelled DRN-projecting RGCs in these animals. It should be noted that anti-Mel-SAP animals in which melanopsin-expressing RGCs were selectively ablated demonstrated escape responses to the looming stimulus similar to VEH controls (VEH $0.94 \pm 0.02$ s versus anti-Mel-SAP $0.91 \pm 0.03$ s, $n = 8$ animals per group; $P > 0.05$) (Supplementary Fig. 5). These data indicate that DRN-projecting RGCs are required for looming-evoked escape responses. In the optomotor test there was no significant difference between VEH and anti-CTB-SAP animals (Fig. 3i), indicating that RGCs mediating this behaviour were unaffected by anti-CTB-SAP treatment and that DRN-projecting RGCs are not required for this behavioural response.

**Looming activation of SC-LP-BLA needs DRN-projecting RGCs.** A subcortical circuit from SC→lateral posterior nucleus of the thalamus→basolateral amygdala (SC-LP-BLA) has been implicated in the regulation of looming-induced freezing in mice, the prominent behaviour observed when looming visual stimuli are presented in an arena that has no shelter[14]. RGCs transmitting looming signals to this circuit cannot be identified using traditional retrograde tracing methods.

To determine whether there is a di-synaptic circuit linking DRN-projecting RGCs to LP, we used a modified rabies transsynaptic tracing method[32]. SC neurons were infected by AAV expressing the RG and histone-Tagged green fluorescent protein (Helper) (Fig. 4a,b). Next, we injected SAD-ΔG-DsRed (EnvA) into LP to infect helper + LP-projecting SC neurons via their presynaptic terminals (Fig. 4c). The double infected rabies-DsRed + /Glyco-GFP + SC relay neurons produce infectious ΔG-rabies-DsRed that propagates transneuronally to infect the RGCs that form synapses with them (Fig. 4c). Double infected neurons were mainly present in the intermediate layer of SC ($n = 6$ animals) (Fig. 4c). Finally, CTB-488 was injected into DRN to determine whether DRN-projecting RGCs could provide di-synaptic input to LP (Fig. 4b). Some of DsRed + RGCs with morphological features of DRN-projecting RGCs were retrogradely labelled by CTB-488 ($n = 127$ cells in 6 animals) (Fig. 4c). This result suggests that DRN-projecting RGCs provide di-synaptic input to LP after synapsing in SC.

To determine whether DRN-projecting RGCs regulate the functional activation of the SC-LP-BLA pathway, a looming-related c-Fos mapping strategy was adopted to assess changes of neuronal activity in the SC-LP-BLA pathway. All animals received DRN CTB injections. Three days later animals received bilateral intraocular injections of 0.1 M PBS ($n = 16$) or anti-CTB-SAP ($n = 8$). Two weeks later all animals were given 3 days of adaptation in the shelterless arena. On the fourth day, a single looming stimulus (15 presentations over 10.75 s) was presented to seven PBS-injected animals (VEH-L) and to the anti-CTB-SAP-treated animals (SAP-L). Under this paradigm VEH-L animals responded by fleeing from the centre of the arena and freezing (Supplementary Movie 2). The remaining PBS-injected animals ($n = 9$) were treated in a similar manner with the exception of the presentation of the looming stimulus (VEH-NL). Looming stimulation significantly increased the number of c-Fos positive neurons in the SC-LP-BLA pathway in VEH-L compared to VEH-NL animals (Fig. 4d–i). The freezing response to the looming stimulus was markedly reduced in SAP-L animals in which DRN-projecting RGCs were ablated (Supplementary Movie 2) and c-Fos induction was significantly reduced (Fig. 4d–i).

In contrast, there was no significant change in the number of c-Fos positive neurons in dorsal periaqueductal grey (dPAG),

which is considered to play a pivotal role in the regulation of defensive behaviour (Fig. 4j,k). These results suggest that DRN-projecting RGCs are part of the SC-LP-BLA circuit that generates the looming-evoked fear response.

**GABA cells activated by looming require DRN-projecting RGCs.** To this point we have established that DRN-projecting RGCs innervate DRN GABA neurons and are required for looming-evoked behaviour. To investigate the response of DRN 5-HT and GABA neurons to looming stimuli, fibre photometry was used to measure $Ca^{2+}$ signals in vivo as a surrogate of neuronal spiking (Fig. 5a). We first confirmed the specificity of the calcium indicator by injecting AAV-DIO-GCaMP6 into the DRN of Sert-Cre or vGAT-Cre mice and examining expression: $>98\%$ of GCaMP6 expression was observed in DRN neurons expressing TPH or GABA in Sert-Cre or vGAT-Cre mice, respectively (Fig. 5b). In freely moving mice, $Ca^{2+}$ signals were recorded from DRN 5-HT or GABA neurons during the presentation of the looming stimulus in an arena without or with a shelter. Each animal was subjected to five looming stimulation trials (10.75 s each) during a 1-h session. In a test arena without a shelter the looming stimulus reliably evoked freezing behaviour accompanied by a decrease in $Ca^{2+}$ signals from 5-HT neurons in Sert-Cre animals (Fig. 5c and Supplementary Fig. 6a,c) and an increase in $Ca^{2+}$ signals from GABA neurons in vGAT-Cre animals (Fig. 5d and Supplementary Fig. 6b,d). These results support the viral tracing experiments suggesting that RGCs transmitting looming signals directly activate DRN GABA interneurons, which inhibit DRN 5-HT neurons (Supplementary Fig. 7). Next, we recorded $Ca^{2+}$ signals from 5-HT neurons when the looming test was conducted in an arena containing a shelter. Under this testing condition $Ca^{2+}$ signals decreased at the beginning of looming stimulation when mice fled to the shelter and subsequently increased dramatically beginning about 4 s later while mice were hiding beneath the shelter (Fig. 5c and Supplementary Fig. 6a). Looming stimulation again produced the opposite responses in GABA neurons, initially significantly increasing $Ca^{2+}$ signals followed by a decrease (Fig. 5d and Supplementary Fig. 6b). These data suggest that once in the safety of the shelter DRN 5-HT neurons are released from GABA inhibition evoked by the looming stimulus. Anti-CTB-SAP treatment eliminated the effects of looming stimulation on $Ca^{2+}$ signals of DRN 5-HT and GABA neurons (Fig. 5e,f and Supplementary Fig. 6a–d). These data support the interpretation that DRN-projecting RGCs are required for looming-evoked behaviour mediated by modulation of DRN 5-HT and GABA neuron activity.

**DRN 5-HT system inhibits looming-evoked defensive responses.** The results provided above suggest that looming stimulation inhibits the activity of DRN 5-HT neurons. We therefore reasoned that activation of DRN 5-HT neurons might attenuate looming-evoked behaviour. To test this hypothesis directly, we first performed long-term specific chemogenetic activation of DRN 5-HT neurons using a Cre-dependent virus encoding the neuronal activator DREADD hM3D (ref. 33). AAV-DIO-hM3D-mCherry was infused into the DRN of Sert-Cre mice (Fig. 6a). Two weeks later, 5-HT neurons were chemogenetically activated via intraperitoneal (i.p.) injection of clozapine N-oxide (CNO) as confirmed by double labelling for mCherry and c-Fos (Fig. 6a). Next, looming-evoked defensive behaviour in an arena with or without a shelter was examined in AAV-DIO-hM3D-mCherry injected Sert-Cre mice injected with saline (hM3D-Saline, $n = 6$) or CNO (hM3D-CNO, $n = 7$) 30 min before behavioural testing. Looming-evoked defensive responses were blocked in animals

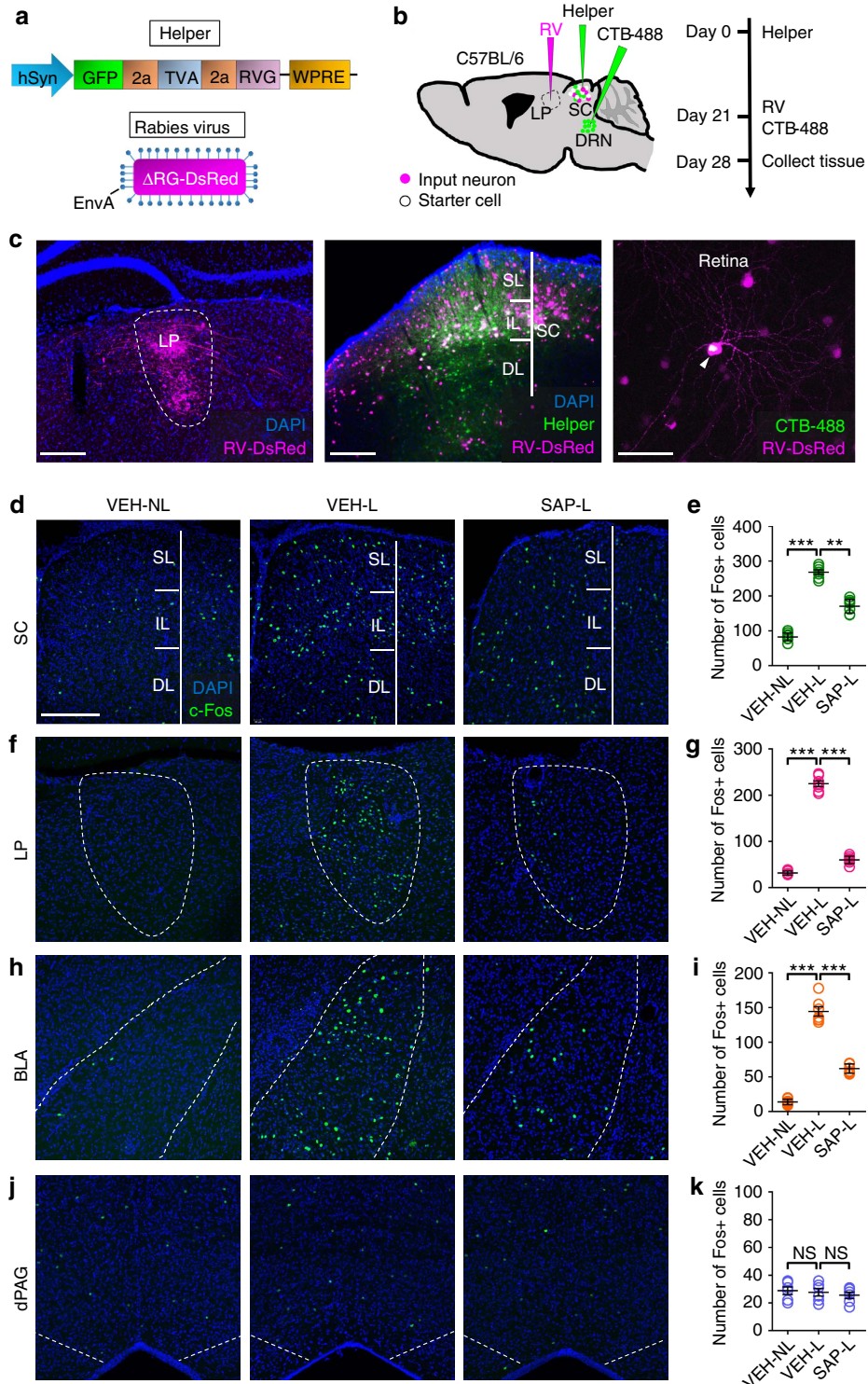

**Figure 4 | DRN-projecting RGCs are necessary for looming-induced c-Fos activation of SC-LP-BLA pathway.** (**a**) Design of helper virus and SAD-ΔG-DsRed (EnvA). (**b**) Experimental design of virus tracing in C57BL/6 mice. (**c**) Left: Injection site of LP. Note there are rabies-DsRed labelled axonal terminals of LP-projecting SC neurons. Middle: SC of a C57BL/6 mouse showing the location of starter cells (white). SL: superficial layer of SC; IL: intermediate layer of SC; DL: deep layer of SC. Right: Retina of a C57BL/6 mouse showing a rabies-DsRed labelled RGC could be co-labelled by CTB-488 deposited into DRN. Arrow head points to the co-labelled RGC. (**d,f,h,j**) Illustration of c-Fos + cells in SC (**d**), LP (**f**), BLA (**h**) and dPAG (**j**) in non-looming stimulated control animals 2 weeks after intraocular injection of PBS (left), looming stimulated animals 2 weeks after intraocular injection of PBS (middle) and looming stimulated animals 2 weeks after intraocular injection of anti-CTB-SAP (right). (**e,g,i,k**) Quantification of c-Fos + cells in SC (**e**), LP (**g**), BLA (**i**) and dPAG (**k**) in each group of animals. One-way ANOVA; **P < 0.001, ***P < 0.0001. Scale bars: (**c**)-left 200 μm; (**c**)-middle 200 μm; (**c**)-right 50 μm; (**d**) 200 μm. Data presented as mean ± s.e.m.

with DRN 5-HT neurons chemogenetically activated (duration from start to end (s): hM3D-Saline, $0.95 \pm 0.21$ versus hM3D-CNO, $10.75 \pm 0$, $P < 0.0001$; peak speed during looming stimulation $(cm\,s^{-1})$: hM3D-Saline, $199.25 \pm 13.25$ versus hM3D-CNO, $91.85 \pm 13.26$, $P < 0.0001$) (Fig. 6c,d and

Supplementary Movie 3). Control Sert-Cre animals ($n = 6$) transfected with AAV-DIO-mCherry and given CNO i.p. responded to the looming stimulus in a manner similar to hM3D-Saline-treated animals demonstrating that CNO itself was not responsible for the inhibition of looming-evoked responses

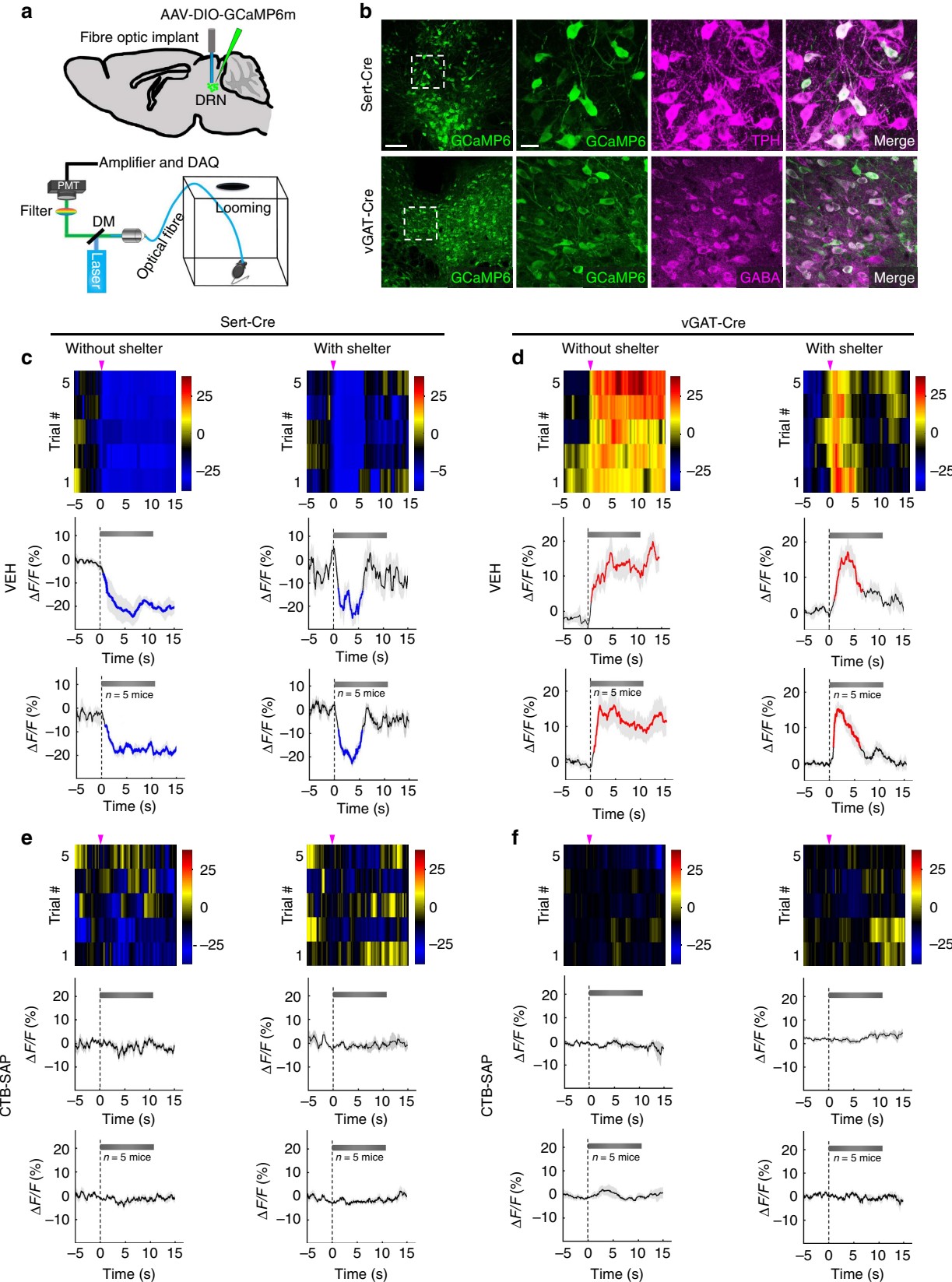

observed in the hM3D-CNO animals (Fig. 6c,d). Next, short-term optogenetic activation of DRN 5-HT neurons during looming stimulation using a Cre-dependent virus encoding channelrhodopsin-2 (ChR2) was performed (Supplementary Fig. 8a–c). Optogenetic activation of DRN 5-HT neurons during looming stimulation also impaired looming-evoked defensive responses (Supplementary Fig. 8d,e and Supplementary Movie 4). Together these results suggest that selective activation of DRN 5-HT neurons inhibits looming-evoked defensive responses.

Since data from *in vivo* fibre photometry recording and *in vitro* brain slice recording suggest that looming signals transmitted by

DRN-projecting RGCs inhibit DRN 5-HT neurons through direct activation of DRN GABA neurons (Fig. 5 and Supplementary Fig. 7). We next asked whether inhibition of DRN GABA neurons could also impair looming-evoked defensive responses. To address this question, we first performed long-term chemogenetic inhibition of DRN GABA neurons using a Cre-dependent virus encoding the inactivating DREADD hM4D (ref. 34). AAV-DIO-hM4D-mCherry was infused into the DRN of vGAT-Cre mice (Fig. 6b). Two weeks later, DRN GABA neurons were chemogenetically inhibited via i.p. injection of CNO as confirmed by double labelling for mCherry and c-Fos (Fig. 6b).

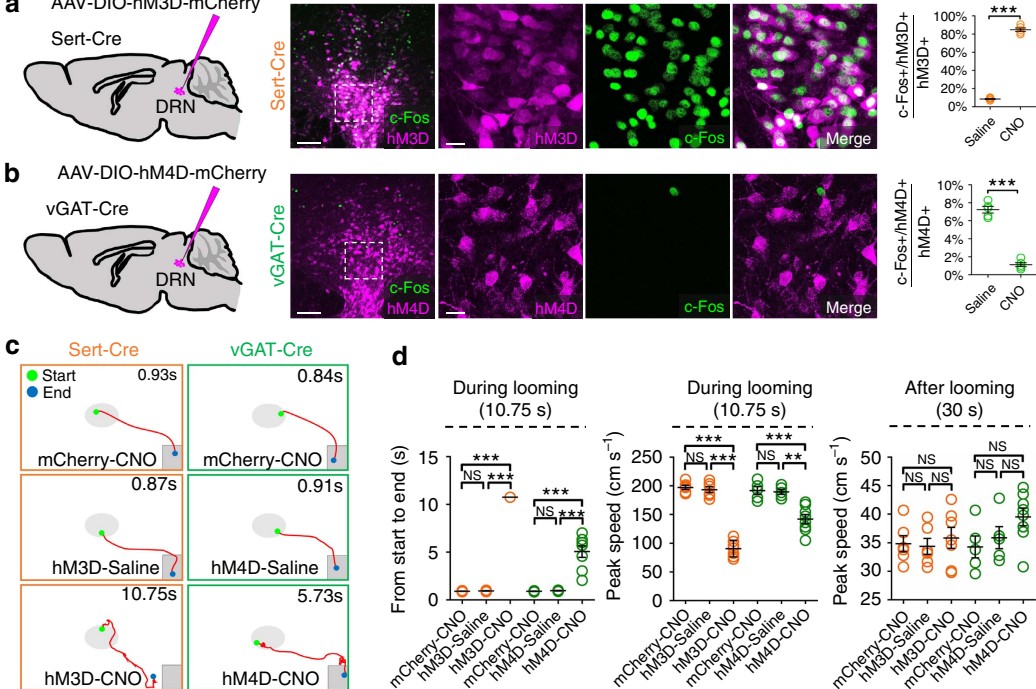

**Figure 6 | Specific activation of DRN 5-HT neurons impairs looming-evoked responses.** (**a**) Left: AAV-DIO-hM3D-mCherry was injected into DRN of Sert-Cre mice. Middle panels: representative images of the DRN illustrating c-Fos in neurons expressing hM3D-mCherry; Right: percentage of total hM3D+ cells expressing c-Fos was significantly increased in animals ($n = 5$) 30 min after CNO administration compared to saline (one-way ANOVA, ***$P < 0.0001$). (**b**) Left: AAV-DIO-hM4D-mCherry was injected into DRN of vGAT-Cre mice. Middle panels: Representative images of the DRN illustrating c-Fos in neurons expressing hM4D-mCherry Right: percentage of total hM4D+ cells expressing c-Fos was significantly decreased in animals ($n = 5$) 30 min after CNO administration compared to saline. One-way ANOVA, ***$P < 0.0001$. (**c**) Representative traces of animal movement during looming stimulation (10.75 s) in CNO-treated controls (mCherry-CNO), saline-treated controls (hM3D-Saline and hM4D-Saline), chemogenetic activation of DRN 5-HT neurons (hM3D-CNO) and chemogenetic inhibition of DRN GABA neurons (hM4D-CNO). (**d**) Duration from start to end points, peak speed during and after looming stimulation in mCherry-CNO, hM3D-Saline and hM3D-CNO-treated Sert-Cre animals (orange), and mCherry-CNO, hM4D-Saline, and hM4D-CNO-treated vGAT-Cre animals (green). One-way ANOVA; **$P < 0.001$; ***$P < 0.0001$; ns = no significant difference. Scale bars: (**a,b**)-left 100 μm; (**a,b**)-right 20 μm. Data presented as mean ± s.e.m.

**Figure 5 | DRN GABA and 5-HT neuron Ca$^{2+}$ signals recorded via fibre photometry during looming stimulation.** (**a**) Schematic of fibre photometry rig used to measure looming-evoked Ca$^{2+}$ signals in freely behaving mice. AAV-DIO-GCaMP6m was injected into DRN of vGAT-Cre or Sert-Cre mice and subsequently a 230 μm fibre optic implant was implanted in the DRN. (**b**) Representative images of DRN from a Sert-Cre and a vGAT-Cre mouse 2 weeks after AAV-DIO-GCaMP6m injection. (**c**) Top: heat maps (five trials) illustrating looming-evoked Ca$^{2+}$ signals of DRN 5-HT neurons from a representative VEH-treated Sert-Cre mouse; Middle: averaged response of the five trials; Bottom: averaged response of the entire test group ($n = 25$ trials from 5 animals) tested in an arena with/without a shelter. Blue segments indicate statistically significant decrease from the baseline ($P < 0.05$; permutation test).
(**d**) Top: heat maps (five trials) illustrating looming-evoked Ca$^{2+}$ signals of DRN GABA neurons from a representative VEH-treated vGAT-Cre mouse; Middle: averaged response of the five trials; Bottom: Averaged response of the entire test group ($n = 25$ trials from 5 animals) tested in an arena with/without a shelter. Red segments indicate statistically significant increase from the baseline ($P < 0.05$; permutation test). (**e**) Top: Heat maps (five trials) illustrating looming-evoked Ca$^{2+}$ signals of DRN 5-HT neurons from a representative Anti-CTB-SAP-treated Sert-Cre mouse; Middle: Averaged response of the five trials; Bottom: Averaged response of the entire test group ($n = 25$ trials from 5 animals) tested in an arena with/without shelter.
(**f**) Top: Heat maps (five trials) illustrating looming-evoked Ca$^{2+}$ signals of DRN GABA neurons from a representative Anti-CTB-SAP-treated vGAT-Cre mouse; Middle: Averaged response of the five trials; Bottom: Averaged response of the entire test group ($n = 25$ trials from 5 animals) tested in an arena with/without shelter. Scale bars: (**b**)-left 100 μm; (**b**)-right 20 μm.

Next, looming-evoked defensive behaviour in an arena with a shelter was examined in vGAT-Cre mice injected with AAV-DIO-hM4D-mCherry 30 min after i.p. injection of saline (hM4D-Saline, $n = 5$) or CNO (hM4D-CNO, $n = 8$). Although looming stimulation continued to evoke escape behaviour in hM4D-CNO-treated animals, inhibition of DRN GABA neurons significantly slowed the response (duration start to end (s): hM4D-Saline, $1.01 \pm 0.26$ versus hM4D-CNO, $5.06 \pm 1.32$, $P < 0.0001$; peak speed during looming stimulation (cm s$^{-1}$): hM4D-Saline, $183.36 \pm 10.53$ versus hM4D-CNO, $147.12 \pm 14.52$, $P < 0.001$) (Fig. 6c,d and Supplementary Movie 5). hM4D-Saline-treated and hM4D-CNO-treated animals were re-tested a week later with a single looming stimulus (0.25 s). The control mice immediately fled to the shelter whereas the CNO-treated animals did not respond to the stimulus (Supplementary Movie 6). Thus using a single trial paradigm CNO-evoked inhibition of DRN GABA neurons blocked the looming response. Control vGAT-Cre animals ($n = 6$) transfected with mCherry-CNO demonstrated rapid looming-evoked responses similar to hM4D-Saline animals (Fig. 6c,d). Next, short-term optogenetic inhibition of DRN GABA neurons during looming stimulation using Cre-dependent virus encoding enhanced halorhodopsin 3.0 (eNpHR3.0) were performed (Supplementary Fig. 8f–h). Optogenetic inhibition of DRN GABA neurons during looming stimulation also impaired looming-evoked defensive responses (Supplementary Fig. 8i,j and Supplementary Movie 7). Thus normal retinally induced activity of DRN GABA neurons is required for the rapid looming-evoked escape response.

In summary, a bistratified RGC type was identified that innervates the DRN and SC via branching axons. Looming signals transmitted by these RGCs result in the activation of the SC-LP-BLA pathway and the indirect inhibition of the DRN serotonergic system via direct activation of DRN GABA interneurons. Inhibition of DRN 5-HT neurons appears to be required to facilitate looming-evoked defensive responses.

## Discussion

The ability to detect and respond to rapidly approaching objects is an innate defensive behaviour fundamental to survival and conserved across species. In mice, it is believed that retinal projections to the SC mediate looming-evoked escape and freezing responses[12–14]. However, because more than 90% of the 30 different identified types of mouse RGC innervate the SC, it has not been possible to use standard retrograde tracers to identify the RGCs that transmit looming-evoked signals centrally[35,36]. In this work, we identified a dedicated circuitry from the retina to the SC and DRN that regulates looming-evoked responses. We began by demonstrating DRN-projecting RGCs in the mouse. This was necessary because it has been reported that unlike other rodents, RGCs did not innervate the mouse DRN. However, the absence of labelled retinal fibres in the mouse DRN was based on anterograde CTB tracing[24] and as recently suggested, this technique underrepresents the retinoraphe projection[37], perhaps due to reduced access of the large CTB molecule at RGC branch points[38]. Here, we demonstrate that RGCs innervate the mouse DRN using retrograde tracing with CTB and importantly, with a modified rabies virus that restricted infection to genetically targeted DRN neurons[26]. We also tested whether DRN/SC-projecting RGCs were capable of responding to a looming stimulus. Extracellular recordings in vitro provided initial evidence that these RGCs could indeed respond to a simulation of a rapidly approaching object. However, additional studies are needed to fully characterize these RGCs including a description of their synaptic inputs.

Using the modified rabies virus retrograde tracing technique we additionally found that DRN-projecting RGCs selectively innervate GABA neurons. It is well documented that DRN GABA neurons play a pivotal role in the modulation of serotonergic output through direct synaptic connections with DRN 5-HT neurons[39–41]. Thus, it seemed reasonable to postulate that activation of the direct retinoraphe pathway would decrease the activity of serotonergic neurons in the DRN. This prediction was borne out by our experiments using fibre photometry to record $Ca^{2+}$ signals from genetically identified DRN 5-HT and GABA neurons in mice during looming stimulation: DRN GABA neurons were activated while the activity of DRN 5-HT neurons was reduced when animals were tested in an arena which contained no shelter. Similar experiments performed in an arena with a shelter were even more informative. In these experiments the looming stimulus reliably induced mice to rapidly flee into the shelter and then freeze for a short period (Supplementary Movie 1). Associated with this behaviour, the $Ca^{2+}$ signals from 5-HT neurons initially decreased at the beginning of looming stimulation when animals fled into the shelter but then increased a few seconds later when the mouse was safely hidden; DRN GABA neurons responded in an opposite manner. We interpret these data based on our previous finding that aversive stimuli activate DRN GABA neurons, but not 5-HT neurons[42,43]. The sense of safety or relief realized after avoiding the threatening stimulus is apparently reflected by activation of DRN 5-HT neurons while DRN GABA neurons are inactivated. In support of these interpretations, looming-stimuli failed to evoke defensive behaviour and had no effect on DRN neuronal activity after selective immunotoxin ablation of DRN-projecting RGCs.

The dual role of 5-HT in the regulation of innate and learned fear responses has long been recognized[44]. Our experiments demonstrated that activation of DRN 5-HT neurons impaired looming-evoked defensive responses. This is consistent with previous findings that 5-HT can play an inhibitory role in the modulation of innate fear responses[20,45,46]. Furthermore, we found that inhibition of DRN GABA neurons significantly retarded the response to the overhead threat. Combined with the in vitro recording data that provide direct evidence that activation of DRN GABA neurons inhibits 5-HT neurons, it appears that looming-induced indirect inhibition of DRN 5-HT neurons is required for the facilitation of looming-evoked defensive responses.

Escape responses to a rapidly approaching overhead predator or to an object on a collision course must be correspondingly rapid and therefore the number of synaptic integrations in the escape circuitry is under temporal constraint. In the current paradigm simulating a rapidly approaching overhead threat, escape responses can be initiated with a latency of less than 250 ms after stimulus onset[7]. However, despite their short latencies, escape responses in animals and humans are made based on an assessment of risk or threat and therefore can be modulated in the short term and in the long term these responses can be modified by learning and memory[47–49]. In the circuitry we describe in the current report, the temporal delays built in by the requirement of the looming stimulus to disinhibit DRN 5-HT neurons must be offset by some advantage. The advantage of the additional synaptic delay may be the plasticity afforded in this circuitry and the widespread projections of the DRN 5-HT neurons. Excitatory inputs directly onto DRN 5-HT neurons from cortical and subcortical sites could play a role in modulating RGC activation of DRN GABA neurons[50].

The looming-evoked visual circuitry we describe expands the role of the DRN in sensory processing by showing that the primary sensory input regulates itself via the DRN. Looming-evoked retinal signals to the DRN are essential to disengage a serotonergic gating mechanism that permits the salient signal to

be processed by the looming circuits to initiate defensive responses. In humans, anxiety or fear might involve a reactivation of innate fear circuits[51]. The inhibitory role of DRN 5-HT neurons in the regulation of looming induced defensive responses may provide an important new mechanism for the usage of selective serotonin reuptake inhibitors in treating fear-related disorders, such as phobia.

## Methods

**Animals.** All experiments were approved by Jinan University Institutional Animal Care and Use Committee. Adult male (6–8 weeks) C57BL/6, Sert-Cre mice (strain name B6.Cg-Tg(Slc6a4-Cre)ET33Gsat; MMRRC, Davis, CA, USA) and vGAT-Cre mice (strain name B6.FVB-Tg(Slc32a1-cre) 2.1Hzo/FrkJ; Jackson Laboratory, Ben Harbor, ME, USA) were used in this study. Animals were housed in a 12 h:12 h light–dark cycle (lights on at 7:00) with food and water provided *ad libitum*. Animals were randomly allocated to experimental and control groups. Experimenters were blind to experimental group and order of testing was counterbalanced during behavioural experiments.

**Surgery and intracranial injection.** Mice were anesthetized (Avertin, 13 µl g$^{-1}$, i.p.) and placed in a stereotaxic instrument (RWD, Shenzhen, China). Erythromycin eye ointment was applied to prevent corneal drying and a heat pad (RWD, Shenzhen, China) was used to hold body temperature at 37 °C. A small craniotomy hole was made using a dental drill (OmniDrill35, WPI) and a micropipette connected to a Quintessential Stereotaxic Injector (Stoelting, Wood Dale, IL, USA) and its controller (Micro4; WPI, Sarasota, USA) was used for injection.

CTB Alexa Fluor conjugates (CTB-488, C-3,477; CTB-594, C-34,777; CTB-647, C-34,778; Invitrogen Inc., Grand Island, NY, USA) was injected into the DRN (0.2 µl per injection, CTB-488 or CTB-647; AP: − 5.2 mm; ML: ± 0 mm; DV: − 2.7 mm, with a 15° angle); intermediate layers of SC (0.4 µl per injection, CTB-594; AP: − 3.7 mm; ML: ± 0.6 mm; DV: − 1.85 mm), followed by 0.03 µl oil (sesame oil; Sigma-Aldrich Corp.) to limit the diffusion of CTB tracer.

For cell-type-specific tracing, a total volume of 0.8 µl containing an equal volume of AAV-EF1α-DIO-EGFP-TVA (2 × 10$^{12}$ particles per ml) and AAV-EF1α-DIO-RVG (2 × 10$^{12}$ particles per ml) was injected at the DRN (eight Sert-Cre mice and eight vGAT-Cre mice) (coordinates: AP: − 5.2 mm; ML: ± 0 mm; DV: − 2.7 mm, with a 15° angle). The pipette was held in place for 10 min, and then withdrawn slowly. Twenty-one days later, 0.4 µl of SAD-ΔG-DsRed (EnvA) (2 × 10$^8$ particles per ml) was injected into the same site.

For di-synaptic tracing retina→SC→LP pathway, 0.8 µl helper virus (rAAV-hSyn-GFP-2a-TVA-2a-RVG-WPRE-pA) (2 × 10$^8$ particles per ml) was injected into the SC (six C57BL/6 mice) (AP: − 3.7 mm; ML: ± 0.6 mm; DV: − 1.85 mm). The pipette was held in place for 10 min, and then withdrawn slowly. Twenty-one days later, 0.4 µl of SAD-ΔG-DsRed (EnvA) (2 × 10$^8$ particles per ml) was injected into the LP (AP: − 2.4 mm; ML: ± 1.5 mm; DV: − 2.2 mm) and 0.2 µl CTB488 were injected into the DRN.

Similar procedures for injection of CTB were adopted to inject 0.4 µl AAV-DIO-GCaMP6m, AAV-DIO-eYFP, AAV-DIO-ChR2-mCherry, AAV-DIO-eNpHR3.0-mCherry and AAV-DIO-mCherry into DRN (virus titres: 4.35 × 10$^{12}$ particles per ml, 10 Sert-Cre mice and 11 vGAT-Cre mice were used for AAV-DIO-GCaMP6m injection; 3 Sert-Cre mice and 3 vGAT-Cre mice were used for AAV-DIO-eYFP injection; 8 Sert-Cre mice and 7 vGAT-Cre mice were used for AAV-DIO-ChR2-mCherry injection; 6 vGAT-Cre mice were used for AAV-DIO-eNpHR3.0- mCherry injection; 10 Sert-Cre mice and 5 vGAT-Cre mice were used for AAV-DIO-mCherry injection).

Similar procedures for injection of CTB were adopted to inject 0.4 µl AAV-DIO-hM3D-mCherry or AAV-DIO-hM4D-mCherry into DRN (virus titres: 2.7 × 10$^{13}$ particles per ml, 35 Sert-Cre mice were used for AAV-DIO-hM3D-mCherry injection; 18 vGAT-Cre mice were used for AAV-DIO-hM4D-mCherry injection).

Following injection, the wound was sutured and antibiotics (bacitracin and neomycin) were applied to the surgical wound and ketoprofen (5 mg kg$^{-1}$) was injected subcutaneously; animals were allowed to recover from anaesthesia under a heat lamp.

**Fibre photometry.** As described previously[43] a fibre photometry system (Thinker Tech, Nanjing) was used for recording Ca$^{2+}$ signals from genetically identified 5-HT and GABA neurons. Briefly, 1 week following AAV-DIO-GCaMP6m virus injection, an optical fibre (230 mm OD, 0.37 numerical aperture (NA); Shanghai Fiblaser) was placed in a ceramic ferrule and inserted towards the DRN through the craniotomy (coordinates: AP: − 5.2 mm; ML: ± 0 mm; DV: − 2.65 mm). The ceramic ferrule was supported with a skull-penetrating M1 screw and dental acrylic. Mice were individually housed and allowed to recover for at least 1 week. To record fluorescence signals, the laser beam from a 488-nm laser (OBIS 488LS; Coherent) was reflected by a dichroic mirror (MD498; Thorlabs), focused by a × 10 objective lens (NA = 0.3; Olympus) and then coupled to an optical commutator (Doric Lenses). An optical fibre (230 mm OD, NA = 0.37, 2-m long)

guided the light between the commutator and the implanted optical fibre. The laser power was adjusted at the tip of optical fibre to the low level of 0.01–0.02 mW, to minimize bleaching.

The GCaMP6m fluorescence was bandpass filtered (MF525-39, Thorlabs) and collected by a photomultiplier tube (R3896, Hamamatsu). An amplifier (C7319, Hamamatsu) was used to convert the photomultiplier tube current output to voltage signals, which was further filtered through a low-pass filter (40 Hz cut-off; Brownlee 440). The analogue voltage signals were digitalized at 500 Hz and recorded by a Power 1401 digitizer and Spike2 software (CED, Cambridge, UK).

**Injection site verification.** After transcardial perfusion with 0.9% saline followed by 4% paraformaldehyde in 0.1 M PBS, the brain was removed and post-fixed with 4% paraformaldehyde overnight at 4 °C, and then transferred into 30% sucrose until sectioning with a cryostat (CM1900, Leica Microsystems, Bannockburn, IL,USA). A series of 40 µm sections were collected for verification of injection sites; tissue was examined under epifluorescence using a Leica, DM6000B microscope.

**Ablation of DRN-projecting RGCs and melanopsin-expressing RGCs.** As described previously[21], C57BL/6 mice received CTB-488 or CTB-647 (for fibre photometry experiments) (0.2 µl peranimal) deposited in the DRN. Three days later, mice received bilateral intraocular injection of a custom conjugated immunotoxin (2 µg per eye) (n = 10) made between saporin (#IT-27-100, Advanced Targeting Systems) and an affinity-purified anti-CTB antibody (C86204M, Meridian Life Science), or anti-Melanopsin-saporin (n = 8) (2 µg per eye) (#IT-44, Advanced Targeting Systems). Mice (n = 10) in the control group received bilateral intraocular injections of an equal volume 0.1 M PBS. After injection, mice were returned to their home cage for 14 days.

**Physiological recording of DRN-projecting RGCs and intracellular injection.** DRN-projecting RGCs were recorded as previously described[21,23]. Briefly, animals were anaesthetized (Avertin, 13 µl g$^{-1}$, i.p.), and eyes were enucleated under dim red light. The lens and vitreous were carefully removed with a pair of fine-forceps. The eyecups were flat-mounted, sclera side down, directly on the bottom of a recording chamber and superfused by oxygenated (95% O$_2$/5% CO$_2$) Ames medium (Sigma-Aldrich, St Louis, MO, USA) at a fixed rate (5 ml min$^{-1}$) at room temperature (22–24 °C). Looming stimuli was generated by programming the Psychophysics Toolbox in Matlab displayed on a Samsung mini LED projector (Samsung SP-P310ME, Samsung Electronics Co Ltd, Suwon City, Korea) and imaged with a first-surface mirror and lens (Edmond Scientific, Barrington, NJ, USA) on the film plane of the microscope's camera port. The luminance level of the projector was measured with a digital radiometer (S370 Radiometer, UDT Instruments, San Diego, CA, USA) using a × 40 water-immersion objective (Carl Zeiss, Thornwood, NY, USA). The irradiance was further reduced using neutral density filters (Oriel Corp., Stratford, CT, USA). Looming responses of DRN-projecting RGCs were recorded using a glass microelectrode and amplified with a patch clamp amplifier (Multiclamp 700B) and digitized (Digidata 1,440; Axon Instrument, Inc., Forest City, CA, USA). RGCs retrogradely labelled by CTB were targeted for recording. Once the physiological profiling was completed, recorded RGCs were intracellularly filled with neurobiotin for morphological evaluation. The acquired physiological data were further analysed off-line (pCLAMP9; Axon Corp., CA, USA).

**Physiological recording from brain slices.** AAV-DIO-ChR2-mCherry or AAV-DIO-eNpHR3.0-mCherry virus was injected into the DRN region of vGAT-Cre or Sert-Cre mice and left for 2–3 weeks. For brain slice preparation, mice were deeply anaesthetized with pentobarbital (100 mg kg$^{-1}$ i.p.) and coronal brain slices were cut at 250 µm thickness. The slices were recovered for 30 min before recording. Whole-cell patch clamp recordings were made on ChR2-negative cell bodies identified under the fluorescence microscope.

The internal solution within whole-cell recording pipettes (3–5 MΩ) contained (in mM): 135 KMeSO$_4$, 10 KCl, 10 HEPES, 10 Na$_2$-phosphocreatine, 4 MgATP, 0.3 Na$_3$GTP (pH 7.2–7.4); or CsMeSO$_4$ 130, NaCl 10, EGTA 10, MgATP 4, Na$_3$GTP 0.3, HEPES 10 (pH 7.2–7.4). 0.5% neurobiotin was included in some recordings for post-recording staining.

Voltage-clamp and current-clamp recordings were performed using a Multiclamp 700B amplifier (Molecular Devices). For voltage-clamp recordings, the neurons were held at − 60 mV. Traces were low-pass filtered at 2 kHz and digitized at 10 kHz (DigiData 1,550, Molecular Devices). The data were acquired and analysed using Clampfit 10.0 software (Molecular Devices). For light stimulation, 470-nm LED or 590-nm LED (Thorlabs) was used to deliver light pulses (blue light: 4 ms × 5 pulses at 20 Hz; yellow light: 1 s) through digital commands from the Digidata 1440. The recorded cells were intracellularly filled with neurobiotin for morphological evaluation.

**Immunocytochemistry.** All animals were anaesthetized (Avertin, 13 µl g$^{-1}$, i.p.) and perfused intracardially with 0.9% saline followed by 4% paraformaldehyde in PBS. Brains and eyes were removed.

For detection of melanopsin, CART and SMI-32 expressing RGCs, retinas were isolated and washed in 0.1 M PBS for three times (10 min each) before incubation in 0.1 M PBS containing 10% normal goat serum (Vector Laboratories, Burlingame, CA, USA) and 0.3% Trition-X-100 (T8787, Sigma-Aldrich, St Louis, MO, USA) for 1 h. Then retinas were incubated for 3 days at 4 °C with a rabbit anti-melanopsin antibody (AB-N38, Advance targeting systems; 1:600) or anti-CART antibody (H-003-62, Phoenix Pharmaceuticals; 1:1,000) or anti-SMI32 antibody (801703, Biolegend; 1:1,000). This was followed by six rinses in 0.1 M PBS and then incubation with a secondary (Dylight 488 or Dylight 549) goat-anti-rabbit IgG (1:400, Vector Laboratories) for 6 h at room temperature.

Retinas with RGCs filled intracellularly with neurobiotin were fixed for 1 h in 4% paraformaldehyde in 0.1 M PBS at room temperature, rinsed in 0.1 M PBS for three times (10 min each) and placed in 10% normal goat serum containing 2% Trition-X-100 for 48 h at 4 °C. Retinas were then incubated in Streptavidin-Alexa Fuor 488 (S32354, Life technologies, 1:100) for 48 h at 4 °C. This was followed by three times rinse in 0.1 M PBS. For labelling of cholinergic amacrine cells, retinas were rinsed in 0.1 M PBS for three times (10 min time $^{-1}$) and placed in 10% normal goat serum containing 2% Trition-X-100 for 1 h at room temperature. Retinas were then incubated in goat-anti-ChAT antibody (AB144P, Millipore; 1:200) for 48 h at 4 °C. This was followed by six times rinse in 0.1 M PBS and then incubation with a secondary antibody Alexa Fluor 594 donkey anti-goat IgG (1:400, A-11058, Molecular Probes) for 6 h at room temperature. Finally, all retinas were rinsed in 0.1 M PBS and cover-slipped in anti-fading aqueous mounting medium (EMS, Hatfield, PA, USA).

For c-Fos labelling, 40 μm cryostat sections containing the DRN, SC, LP, BLA or dorsal periaqueductal grey were placed in blocking solution for 1 h before incubation in primary antibody against c-Fos (rabbit, 1:500; PC38T, Calbiochem) (36 h at 4 °C). Sections were then incubated with corresponding secondary antibody at a dilution of 1:400 for 6 h at room temperature: goat anti-rabbit Alexa 488(107909, Jackson ImmunoResearch).

For TPH or GABA labelling, procedures used in c-Fos labelling were adopted except that the primary antibody was replaced by anti-TPH (mouse, TPH; 1:1,000; T8575, Sigma-Aldrich) or anti-GABA (mouse, 1:1,000; A0310, Sigma-Aldrich), and the secondary antibody was replaced by goat anti-mouse Alexa 594 (115-587-003, Jackson ImmunoResearch) or goat anti-mouse Alexa 647 (115-587-003, Jackson ImmunoResearch).

Finally, all brain sections were rinsed in 0.1 M PBS and cover-slipped in anti-fading aqueous mounting medium with DAPI (EMS, Hatfield, PA, USA).

**Image analysis.** Retinas and sections were imaged with a Zeiss 700 confocal microscope with ×5 or ×20 objectives, or a ×40 oil immersion objective. For three-dimensional reconstruction of injected or virus labelled RGCs, optical sections were collected at 0.2 μm intervals. Each stack of optical sections covered a retinal area of 325.75 × 325.75 mm$^2$ (1,024 × 1,024 pixels). Using Image J and Photoshop CS5 (Adobe Corp., San Jose, CA, USA), each stack of optical sections was montaged and projected to a 0° X–Y plane and a 90° Y–Z plane to obtain a three-dimensional reconstruction of the cell. Details of three-dimensional reconstruction and confocal calibration procedures were described elsewhere[21]. Contrast and brightness were adjusted and the red-green images had been converted to magenta-green. Total soma and dendritic field size of each filled cell were analysed. Dendritic field area was calculated by drawing a convex polygon linking the dendritic terminals. The dendritic field area was then calculated and the diameter expressed as that of a circle having an equal area.

**Looming stimulation test.** The looming stimulation test was performed in a 50 × 50 × 37 cm closed arena with or without a shelter. An LED monitor was embedded into the ceiling to present the looming stimulus. The looming stimulus, which consisted of an expanding black disc, appeared at a diameter of 2° to 20° visual angle in 0.25 s, and was presented once in 0.25 s or 15 times in 10.75 s. The stimulation was triggered by the experimenter manually when mouse was in the centre of the arena.

For fibre photometry recording of Ca$^{2+}$ signals from genetically identified DRN 5-HT and GABA neurons, an implanted optic fibre was connected to the fibre photometry system, and the Ca$^{2+}$ signals were recorded during the whole test session (looming stimulus was presented 15 times per trial).

For the chemico-genetic experiments, Sert-Cre or vGAT-Cre mice were intraperitoneally treated with either saline or Clozapine N-oxide (CNO, 2 mg kg$^{-1}$, ab141704, abcam) 30 min before testing in an arena with or without a shelter (Looming stimulus was presented 15 times per trial).

For the optogenetic activation experiment, an implanted fibre optic was connected to a 463-nm blue light laser or 590-nm yellow light laser and the light power was set to about 10 mW at the fibre tip. The duration of neuronal activation fully encompassed the duration of the looming stimulus.

**Optomotor test.** Optomotor responses of animals in VEH ($n = 6$ animals) and anti-CTB-SAP ($n = 6$ animals) groups were measured as described by Abdeljalil et al.[52] Briefly, mice were placed on a platform in the form of a grid (12 cm diameter, 19.0 cm above the bottom of the drum) surrounded by a motorized drum (29.0 cm diameter) that could be revolved clock-wise or anticlockwise at two

revolutions per minute. After 10 min of adaptation in the dark, vertical black and white stripes of a defined spatial frequency were presented to the animal. These stripes were rotated alternately clockwise and anticlockwise, for 2 min in each direction with an interval of 30 s between the two rotations. Various spatial frequencies subtending 0.03, 0.13, 0.26, 0.52 and 1.25 cycles degree$^{-1}$ were tested individually on different days in a random sequence. Animals were videotaped with an infrared digital video camera for subsequent scoring of head tracking movements. Procedures for measuring optomotor responses under photopic condition were similar to the scotopic condition except that animals were subjected to 400 Lux during 5 min to allow them to adapt to the light.

**Statistics.** Data analysis was done by experimenters blind to experimental conditions. One-way ANOVA were used to quantify duration from start to end, peak speed during and after looming stimulation, number of c-Fos + cells, and ipRGCs. Data are shown as mean ± s.e. of the mean (s.e.m.). Statistical significance was set at $P < 0.05$.

Multivariate permutation test was used to analyse the statistical significance of the event-related fluorescence change (ERF). One thousand permutations for an a-level of 0.05 were used to compare the values of $\Delta F/F$ at each time point with the ERF baseline values. A series of statistical $P$ value at each time point were generated and the statistical results were superimposed on the average ERF curve with red and blue lines indicating statistically significant ($P < 0.05$) increase or decrease, respectively.

To calculate the amplitude and activation duration of $\Delta F/F$ values, we first segmented the data based on the behavioural events of with/without looming stimulation. The 95% fraction of the baseline $\Delta F/F$ values defined as the up-threshold value and the 5% fraction as the low threshold. We then detected the local signal peaks during a given interaction bout using the MATLAB findpeaks function. The response amplitude of a behavioural bout was calculated by averaging all the local peaks above the up-threshold value. The activation duration and inhibition duration was calculated by summing the time points above the up threshold or below the low threshold, respectively. The average bout peaks, activation duration and inhibition duration were averaged across the behavioural session to report the corresponding values of a given mouse.

**Data availability.** Data from the experiments presented in the current study are available from the corresponding authors on reasonable request.

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

## Acknowledgements

We would like to thank Dr. Wenbiao Gan for comments, Dr Xin Huang for his help in writing custom software for looming stimulation, and Dr Xiaobin He and Dr Sen Jin for providing technical support. We also thank Ms Ruanna Wang for her help in genotyping. This work was supported by grants from the National Natural Science Foundation of China (31400942, 81601969 and 31571091), the National Program on Key Basic Research Project of China (973 Program, 2014CB542205), Guangdong Natural Science Foundation (2014A030313387 and S2013040014831), funds of Leading Talents of Guangdong (2013), Programme of Introducing Talents of Discipline to Universities (B14036), National Basic Research program (2015CB351800), the Fundamental Research Funds for the Central Universities Grant (21609101) and the USA National Institutes of Health R01 NS077003.

## Author contributions

Conceived and designed the experiments, C.R., K.-F.S., G.E.P.; Performed the experiments, L.H., T.Y., M.T., Z.Z., Y.X., Y. Hu, Q.T., Y. Han, J.Z., C.R; Analysed the data, C.R., L.H., T.Y., G.E.P.; Contributed reagents/materials/analysis tools: C.R., K.-F.S., F.X., M.L., M.P., P.J.S. and G.E.P.; Writing—review and editing, C.R., G.E.P., K.-F.S., P.J.S.; Funding acquisition, C.R., K.-F.S., Q.T., M.P. and G.E.P.; Supervision, C.R.

## Additional information

**Competing interests:** The authors declare no competing financial interests.

