## [Peer Review File · Nature Communications]

Reviewers' expertise:

Reviewer #1: Visual processing;

Reviewer #2: Innate fear circuit;

Reviewer #3: 5-HT system in behaviors.

Reviewers' comments:

Reviewer #1 (Remarks to the Author):

This is an impressive study that reveals DRN-projecting RGCs in mice and their roles in looming-evoked defensive response. The authors discovered that these RGCs project to SC and GABAergic neurons in the DRN, they are necessary for looming-evoked responses, and manipulating DRN activity can affect looming-evoked responses. These findings should be of great interests to many who study visual behaviors and those who study aversive and affective behaviors. Overall, the experiments are well designed and clearly presented. The data are of high quality and are extensive. I do not have any major issue with this manuscript, but an obvious missing piece is a characterization of these DRN-projecting RGCs. The authors showed that these cells were not stained by melanopsin, CART or SMI-32. But are they positive for any known markers? Are they On, Off or On-Off cells (the dendritic laminar patterns suggest they are On-Off)? What visual stimuli do they respond to? I am not asking the authors to provide a detailed functional study of these cells. Frankly, with such data, this study could be published in a journal with an even higher profile. But any functional data of these cells would make this paper even more impactful. Short of such data, maybe some speculation of how these cells could be mediating looming response would help.

Reviewer #2 (Remarks to the Author):

This is a timely nice study evaluating the contribution of a retino-raphé neuronal pathway in mediating defensive behavior in response to a looming stimulus. The authors combine an impressive variety of approaches from behavior to slice physiology, cell specific retrograde tracing, pharmacogenetic and optogenetic. In a first step, using retrograde tracing and rabies-based transynaptic retrograde labeling, the authors identified a direct anatomical connection between retinal ganglion cells (RGC) and the dorsal raphe nucleus (DRN) and the presence of collaterals innervating the superior colliculus (SC). Next, using a specific ablation of DRN-projecting RGCs, they demonstrate the necessity of this pathway in mediating looming-evoked defensive responses. In a follow up experiment, the authors also demonstrate that SC neurons receiving collaterals from the DRN-projecting RGC cells project to the thalamus and that the specific ablation of DRN-projecting RGCs reduced neuronal activation in the basolateral amygdala associated with looming-evoked responses. Using photometry and slice recordings they demonstrate that RGC directly contact GABAergic neurons in the DRN, which inhibit 5HT DRN neurons. Accordingly, pharmacogenetic activation of 5HT DRN neurons or inhibition of GABAergic DRN cells blocked looming evoked responses. In a last experiment, the authors observed calcium transient in the BLA during looming-evoked defensive responses that were abolished by the chemogenetic inactivation of 5HT DRN neurons. The authors concluded that they have identified a novel retino-raphé pathway mediating looming-evoked defensive responses.

The combination of approaches used by the authors is very impressive and allowed them to unambiguously demonstrate a role of DRN 5HT neurons in looming-evoked neuronal responses. However, several issues dampened my enthusiasm for the manuscript and are related to behavioral experiments and to some conceptual issues that need to be addressed.

1) The first concern I have is related to the narrative of the manuscript and especially the last section in which the authors try to make the point that DRN 5HT neurons somehow influence the amygdala to promote looming-evoked responses (Figure 6e, f). However the experimental evidence related to this section is rather weak. I think this section will deserve a full story and should be removed from the present version of the manuscript. Below, the authors will find my arguments why I think this section is absolutely not convincing and should be removed. During looming-evoked defensive responses, DRN 5HT neurons are inhibited by DRN GABAergic neurons whereas amygdala neurons seem activated (Figure 6f). How these two findings match together is rather unclear? Many possibilities could account for the modulation of amygdala neurons by DRN 5HT neurons, direct projections to the amygdala or alternatively projection to the SC. Another possibility will be a direct modulation of avoidance centers in the dIPAG, which are known to receive direct inputs from DRN 5HT neurons. None of these possibilities are fully explored in the present version of the manuscript, which is mandatory if the authors want to follow up with the current version of the manuscript. Alternatively, I suggest the authors to focus their paper on their convincing demonstration that DRN 5HT neurons mediate looming-evoked defensive responses.

2) A second aspect of the manuscript deserving a deeper investigation is the photometry section and particularly how behavior is modulated accordingly to calcium changes, something, which is not clear from the present version of the manuscript. For instance in figure 5 panel c-f, it is not clear what the mice are actually doing during the five trials, do they freeze, do they move, are they under the shelter? More particularly, how locomotion is correlated to changes in calcium imaging is missing. The authors should provide the behavioral data and variability for individual animals for each of the 5 trials. Moreover,

3) I have an issue with the baseline calcium signal in fig 5 c bottom, in which there is a net increase in calcium signals just before the trial onset, which biased, the supposed inhibition observed later on. There is absolutely no quantification of these data, something that must be done in a revised version of the manuscript. On a related note, during ablation of DRN-projecting RGC, it is not clear if this manipulation induces changes in the speed of the animals. There is still a possibility that these animals are afraid by the looming stimulus but could not detect the shelter as a safe place?

4) In figure 5, panel d left, it seems there is a mistake with the scale of the top and bottom graphs. In the top graph, the signal is clearly oversaturated (at 20 on the scale for all 5 trials) after the trial onset but this is not reflected in the averaged bottom graph with an average around 10 after trial onset. This should be clarified and the scales verified. Also on the bottom graphs, the green area is supposed to represent time under the shelter but there is no variability: how does this relate to individual trials?

5) I have also some concerns related to figure 4, c-fos experiment section. In this experiment the authors clearly evaluate c-fos expression in areas known to be involved in looming-evoked responses. Additional controls should be provided here. In particular it is not clear if the ablation of DRN-projecting RGC neurons reduces c-fos expression in structures not supposed to be involved in defensive reaction. Moreover, what about the dorsal PAG, which received direct 5HT, inputs emanating from the DRN and which is involved in fear avoidance behavior. Additional analyses are required here.

6) Quantification of morphological similarities between CTB labeled-DRN projecting cells and rabies labeled DRN projecting cells should be provided (eccentricity measure for instance).

7) It is not clear why optogenetic experiments were performed only for 5HT DRN neurons and not VGAT DRN neurons. These experiments should be provided as well to strengthen the overall impact of the manuscript.

Reviewer #3 (Remarks to the Author):

Huang et al. studied a projection from retinal ganglion cells to the dorsal raphe, which also project to superior colliculus. They showed that this projection is involved in behavioral responses to looming visual stimuli in mice, suggesting that it could be specialized for defensive behavior. This is an interesting study. Unfortunately, there are important control experiments missing, and it is difficult to evaluate certain conclusions.

Major comments:

1. The first control experiment missing is photometry from EYFP+ neurons to measure potential movement artifact.
2. The second control experiment missing is CNO injections in WT mice, to show that the chemogenetic effects are not due to drug itself.
3. There are many techniques used here, and there is insufficient methodological description to evaluate the data. For the photometry measurements, how was dF/F calculated? The viral injection sites look very small. How was this achieved with such large injection volumes (0.4 or 0.2 μ l)?

Minor comments:

1. The color scales for the photometry measurements are misleading due to color transition nonlinearities. For example, red regions appear categorically different from yellow ones, although the quantitative differences between those colors may be small. Because the measure (dF/F) is a nonnegative number, I suggest changing the scales to a single gradient (e.g., black to white).
2. The title is a bit misleading, as it suggests that the projection to the DR regulates looming-defense behavior. The authors did not distinguish this projection from the collaterals to SC.

Response to Referees Letter

Reviewer #1 (Remarks to the Author):

This is an impressive study that reveals DRN-projecting RGCs in mice and their roles in looming-evoked defensive response. The authors discovered that these RGCs project to SC and GABAergic neurons in the DRN, they are necessary for looming-evoked responses, and manipulating DRN activity can affect looming-evoked responses. These findings should be of great interests to many who study visual behaviors and those who study aversive and affective behaviors. Overall, the experiments are well designed and clearly presented. The data are of high quality and are extensive. I do not have any major issue with this manuscript, but an obvious missing piece is a characterization of these DRN-projecting RGCs. The authors showed that these cells were not stained by melanopsin, CART or SMI-32. But are they positive for any known markers? Are they On, Off or On-Off cells (the dendritic laminar patterns suggest they are On-Off)? What visual stimuli do they respond to? I am not asking the authors to provide a detailed functional study of these cells. Frankly, with such data, this study could be published in a journal with an even higher profile. But any functional data of these cells would make this paper even more impactful. Short of such data, maybe some speculation of how these cells could be mediating looming response would help.

A: Reviewer #1 was very positive about our study and had only a single suggestion which was to provide characterization of the DRN-projecting RGCs. As the reviewer stated “I am not asking the authors to provide a detailed functional study of these cells”. In response, we have performed an initial functional characterization of DRN-projecting RGCs as suggested by the reviewer by recording extracellular responses of CTB-labeled DRN-projecting RGCs in a retinal whole mount maintained *in vitro*. We used a looming stimulus similar to the one used for the *in vivo* experiments—a rapidly expanding dark disk presented on a grey background. All of

the 11 recorded DRN-projecting RGCs showed a transient increase in their firing rate at the onset of the dark looming stimulus. This preliminary characterization provides the necessary proof-of-principle that DRN-projecting RGCs are capable of responding to a looming stimulus. The data have been added to Figure 3 (new Fig. 3a). There is clearly much more to be learned about these RGCs but we strongly feel that additional functional characterization of these cells requires whole-cell patch-clamp recording and is beyond the scope of this initial report. (Article File, page 7, line 155-162)

Reviewer #2 (Remarks to the Author):

This is a timely nice study evaluating the contribution of a retino-raphé neuronal pathway in mediating defensive behavior in response to a looming stimulus. The authors combine an impressive variety of approaches from behavior to slice physiology, cell specific retrograde tracing, pharmacogenetic and optogenetic. In a first step, using retrograde tracing and rabies-based transynaptic retrograde labeling, the authors identified a direct anatomical connection between retinal ganglion cells (RGC) and the dorsal raphe nucleus (DRN) and the presence of collaterals innervating the superior colliculus (SC). Next, using a specific ablation of DRN-projecting RGCs, they demonstrate the necessity of this pathway in mediating looming-evoked defensive responses. In a follow up experiment, the authors also demonstrate that SC neurons receiving collaterals from the DRN-projecting RGC cells project to the thalamus and that the specific ablation of DRN-projecting RGCs reduced neuronal activation in the basolateral amygdala associated with looming-evoked responses. Using photometry and slice recordings they demonstrate that RGC directly contact GABAergic neurons in the DRN, which inhibit 5HT DRN neurons. Accordingly, pharmacogenetic activation of 5HT DRN neurons or inhibition of GABAergic DRN cells blocked looming evoked responses. In a last experiment, the authors observed calcium transient in the BLA during looming-evoked defensive responses that were abolished by the chemogenetic inactivation of 5HT DRN neurons. The authors concluded that they have identified a novel retino-raphé pathway mediating looming-evoked defensive responses.

The combination of approaches used by the authors is very impressive and allowed them to unambiguously demonstrate a role of DRN 5HT neurons in looming-evoked neuronal responses. However, several issues dampened my enthusiasm for the manuscript and are related to behavioral experiments and to some conceptual issues that need to be addressed.

1) The first concern I have is related to the narrative of the manuscript and especially the last section in which the authors try to make the point that DRN 5HT neurons somehow influence the amygdala to promote looming-evoked responses (Figure 6e, f). However the experimental evidence related to this section is rather weak. I think this section will deserve a full story and should be removed from the present version of the manuscript. Below, the authors will find my arguments why I think this section is absolutely not convincing and should be removed. During looming-evoked defensive responses, DRN 5HT neurons are inhibited by DRN GABAergic neurons whereas amygdala neurons seem activated (Figure 6f). How these two findings match together is rather unclear? Many possibilities could account for the modulation of amygdala neurons by DRN 5HT neurons, direct projections to the amygdala or alternatively projection to the SC. Another possibility will be a direct modulation of avoidance centers in the dIPAG, which are known to receive direct inputs from DRN 5HT neurons. None of these possibilities are fully explored in the present version of the manuscript, which is mandatory if the authors want to follow up with the current version of the manuscript. Alternatively, I suggest the authors to focus their paper on their convincing demonstration that DRN 5HT neurons mediate looming-evoked defensive responses.

A: Reviewer #2 was also overall very positive about our findings but indicated some concerns. Reviewer #2 indicates that we have ‘unambiguously’ demonstrated a role for DRN 5HT neurons in looming-evoked neuronal responses and in comment 1) suggests we focus our paper on these results and not include the BLA data presented

in Figure 6e and 6f as this aspect of the data set is not as fully explored as other components of the paper. In response to the reviewer's suggestion we have removed this experiment from the revised manuscript and in so doing have deleted Figures 6e and 6f. We have also greatly reduced the discussion regarding the BLA.

2) A second aspect of the manuscript deserving a deeper investigation is the photometry section and particularly how behavior is modulated accordingly to calcium changes, something, which is not clear from the present version of the manuscript. For instance in figure 5 panel c-f, it is not clear what the mice are actually doing during the five trials, do they freeze, do they move, are they under the shelter? More particularly, how locomotion is correlated to changes in calcium imaging is missing. The authors should provide the behavioral data and variability for individual animals for each of the 5 trials.

A: Here the reviewer has asked for a more detailed description of the photometry data and the relationship between mouse behavior and the calcium imaging data presented. In response to this request we have included the representative raw traces of GCaMP fluorescence changes to looming stimulation and an indication of the general behavior of the mice (exploring/fleeing or rapidly escaping/freezing) that was extracted from the video recordings (new Supplementary Fig. 6a,b). We also added the analysis of correlation between speed and calcium signals which indicated that locomotion did not affect neuronal activity of DRN 5-HT or GABA neurons (new Supplementary Fig. 6c,d).

3) I have an issue with the baseline calcium signal in fig 5 c bottom, in which there is a net increase in calcium signals just before the trial onset, which biased, the supposed inhibition observed later on. There is absolutely no quantification of these data, something that must be done in a revised version of the manuscript. On a related note, during ablation of DRN-projecting RGC, it is not clear if this manipulation induces changes in the speed of the animals. There is still a possibility that these animals are

afraid by the looming stimulus but could not detect the shelter as a safe place?

A: We have re-analyzed the fiber photometry data and replaced the original Fig. 5c with new Figure more representative of the entire data set. We also added the quantification of GCaMP fluorescence changes from the entire test group to new Fig. 5c-f.

The reviewer raises a question about ‘the speed of the animals’ following DRN-projecting RGC ablation. Kinetic analyses (speed as cm/s) are presented in new Fig. 3h where it is clear that animals with RGCs ablated do not run from the looming stimulus. The reviewer asks whether the looming stimulus may have evoked a fear response but perhaps the animals ‘could not detect the shelter as a safe place’. As illustrated in new Fig. 3g, the mice continue to explore the arena during the looming stimulus. When animals do not have a shelter in which to hide they freeze in response to the looming stimulus. It is very clear from new Fig. 3g and Supplementary Movie 2 that the animals do not freeze which might be expected if they were unaware that a shelter existed. Moreover, data presented in new Fig. 3i shows that an analysis of another visually-evoked behavior, the optomotor response, was not altered by ablation of DRN-projecting RGCs. We are convinced that the animals are aware that there is a shelter but do not flee to the shelter because they are unaware of the looming stimulus.

4) In figure 5, panel d left, it seems there is a mistake with the scale of the top and bottom graphs. In the top graph, the signal is clearly oversaturated (at 20 on the scale for all 5 trials) after the trial onset but this is not reflected in the averaged bottom graph with an average around 10 after trial onset. This should be clarified and the scales verified. Also on the bottom graphs, the green area is supposed to represent time under the shelter but there is no variability: how does this relate to individual trials?

A: We thank the reviewer for noticing this error and the scales in Fig. 5d have been

corrected.

We agree that the green areas in Fig. 5 were not accurate. We removed the green areas in the new Fig. 5 and added the representative raw traces of GCaMP fluorescence changes to looming stimulation and an indication of the general behavior of the mice (exploring/fleeing or rapidly escaping/freezing) that was extracted from the video recordings (new Supplementary Fig. 6a,b).

5) I have also some concerns related to figure 4, cfos experiment section. In this experiment the authors clearly evaluate cfos expression in areas known to be involved in looming-evoked responses. Additional controls should be provided here. In particular it is not clear if the ablation of DRN-projecting RGC neurons reduces c-fos expression in structures not supposed to be involved in defensive reaction. Moreover, what about the dorsal PAG, which received direct 5HT, inputs emanating from the DRN and which is involved in fear avoidance behavior. Additional analyses are required here.

A: In the revised manuscript, we added the analysis of c-Fos expression in dorsal PAG. We found there was no significant difference of c-Fos expression in dPAG between VEH-NL, VEH-L and SAP-L groups (new Fig. 4j,k). These data are consistent with the work of Wei and colleagues¹, which suggested that dPAG was not involved in the regulation of looming induced defensive behavior. (Article File, page 11, line 238-240)

6) Quantification of morphological similarities between CTB labeled-DRN projecting cells and rabies labeled DRN projecting cells should be provided (eccentricity measure for instance).

A: The reviewer requests additional quantification of the similarities between CTB-labeled DRN-projecting RGCs and rabies virus labeled DRN-projecting RGCs. Comparison of diameters of soma and dendritic field were added to Fig 2 (new Fig.

2g). Eccentricity measure of rabies labeled DRN-projecting RGCs was added to Supplementary Fig 4 (new Supplementary Fig. 4c). (Article File, page 7, line 146-148)

7) It is not clear why optogenetic experiments were performed only for 5HT DRN neurons and not VGAT DRN neurons. These experiments should be provided as well to strengthen the overall impact of the manuscript.

A: Additional optogenetic data from vGAT DRN neurons has been provided as requested and is presented in new Supplementary Fig. 8f-j and Supplementary Movie. 7. (Article File, page 15, line 326-331)

Reviewer #3 (Remarks to the Author):

Huang et al. studied a projection from retinal ganglion cells to the dorsal raphe, which also project to superior colliculus. They showed that this projection is involved in behavioral responses to looming visual stimuli in mice, suggesting that it could be specialized for defensive behavior. This is an interesting study. Unfortunately, there are important control experiments missing, and it is difficult to evaluate certain conclusions.

Major comments:

1. The first control experiment missing is photometry from EYFP+ neurons to measure potential movement artifact.

A: Measurement of potential movement artifact from DRN 5-HT/GABA neurons transfected with AAV-DIO-EYFP was added to new Supplementary Fig. 6a,b.

2. The second control experiment missing is CNO injections in WT mice, to show that the chemogenetic effects are not due to drug itself.

A: To exclude the possibility that CNO treatment might influence the result of looming induced defensive responses, we first injected AAV-DIO-mCherry into DRN of Sert-Cre and vGAT-Cre mice. 2 weeks later we treated those mice with CNO (mCherry-CNO group). Looming test was conducted 30 mins after CNO treatment. As illustrated in new Fig. 6c,d, there was no significant difference in the comparison of kinetic parameters between mCherry-CNO and hM3D/hM4D-Saline groups. (Article File, page 13-15, line 292-296, line 324-326)

3. There are many techniques used here, and there is insufficient methodological description to evaluate the data. For the photometry measurements, how was dF/F calculated? The viral injection sites look very small. How was this achieved with such large injection volumes (0.4 or 0.2 μ l)?

A: The description of calculation of dF/F was added to the methods part (Statistics section). (Article File, page 30, line 649-659)

CTB was injected followed by 0.03 μ l sesame oil, which has been shown to limit the diffusion of CTB tracer². We have added this to the Methods. (Article File, page 20, line 435-440)

The beauty of the G-deleted rabies virus labeling technique used in combination with Cre-recombinase helper viruses and genetic mouse models (i.e., selective expression of Cre-recombinase) is that the functional injection sites are constrained by the requirement that rabies virus infection is restricted to only cells expressing the helper virus and helper viruses only replicate in neurons expressing Cre-recombinase. Therefore, the volume of injected virus is not the determining factor in the size of the injection site (i.e., the number and distribution of 'starter' cells). This is quite different from standard tracer injections in which the volume of injected tracer and its ability to diffuse through the brain parenchyma does determine the size of the injection site.

Minor comments:

1. The color scales for the photometry measurements are misleading due to color transition nonlinearities. For example, red regions appear categorically different from yellow ones, although the quantitative differences between those colors may be small. Because the measure (dF/F) is a nonnegative number, I suggest changing the scales to a single gradient (e.g., black to white).

A: In the new Fig. 5, the color scales for the photometry measurements were changed to blue to red, which has been used in our previous work³.

2. The title is a bit misleading, as it suggests that the projection to the DR regulates looming-defense behavior. The authors did not distinguish this projection from the collaterals to SC.

A: We are in agreement with the notion that the retinal input to the SC is responsible for activating the circuits that either initiate the escape behavior or freezing in response to looming stimuli. However, the same RGCs must alter DRN serotonergic activity for the behavioral responses to be expressed. Thus, the retinal projection to the DRN regulates the behavioral response but does not drive it. In response to the reviewer we have rearranged the wording of the title “A retinoraphe projection regulates serotonergic activity and looming-evoked defensive behavior”. (Article File, page 1, line 1-3)

References

1. Wei, P. *et al.* Processing of visually evoked innate fear by a non-canonical thalamic pathway. *Nat Commun* **6**, 6756 (2015).
2. Lee, H.S., Park, S.H., Song, W.C. & Waterhouse, B.D. Retrograde study of hypocretin-1 (orexin-A) projections to subdivisions of the dorsal raphe nucleus in the rat. *Brain Res* **1059**, 35–45 (2005).
3. Li, Y. *et al.* Serotonin neurons in the dorsal raphe nucleus encode reward signals. *Nat Commun* **7**, 10503 (2016).

REVIEWERS' COMMENTS:

Reviewer #1 (Remarks to the Author):

The authors have fully addressed my comment.

Reviewer #2 (Remarks to the Author):

The authors have performed a number of control experiments to address the points I have raised. These controls have substantially increased the overall impact of the paper. I don't have any additional comments on this beautiful piece of work. I recommend publication of this manuscript as it is.

Reviewer #3 (Remarks to the Author):

The authors have addressed my concerns.

Response to Referees Letter

Reviewer #1 (Remarks to the Author):

The authors have fully addressed my comment.

A: We thank the reviewer for the positive comments.

.

Reviewer #2 (Remarks to the Author):

The authors have performed a number of control experiments to address the points I have raised. These controls have substantially increased the overall impact of the paper. I don't have any additional comments on this beautiful piece of work. I recommend publication of this manuscript as it is.

A: We thank the reviewer for the positive comments.

Reviewer #3 (Remarks to the Author):

The authors have addressed my concerns

A: We thank the reviewer for the positive comments.